# Understanding Sensitivity of Differential Attention through the Lens of Adversarial Robustness

**Tsubasa Takahashi**[1*]  **Shojiro Yamabe**[2†]  **Futa Waseda**[3†]  **Kento Sasaki**[1,4]
[1]Turing Inc.  [2]Institute of Science Tokyo  [3]The University of Tokyo  [4]University of Tsukuba
tsubasa.takahashi@acm.org, kento.sasaki@turing-motors.com

## ABSTRACT

Differential Attention (DA) has been proposed as a refinement to standard attention, suppressing redundant or noisy context through a subtractive structure and thereby reducing contextual hallucination. While this design sharpens task-relevant focus, we show that it also introduces a structural fragility under adversarial perturbations. Our theoretical analysis identifies *negative gradient alignment*—a configuration encouraged by DA's subtraction—as the key driver of sensitivity amplification, leading to increased gradient norms and elevated local Lipschitz constants. We empirically validate this Fragile Principle through systematic experiments on ViT/DiffViT and evaluations of pretrained CLIP/DiffCLIP, spanning five datasets in total. These results demonstrate higher attack success rates, frequent gradient opposition, and stronger local sensitivity compared to standard attention. Furthermore, depth-dependent experiments reveal a robustness crossover: stacking DA layers attenuates small perturbations via *depth-dependent noise cancellation*, though this protection fades under larger attack budgets. Overall, our findings uncover a fundamental trade-off: DA improves discriminative focus on clean inputs but increases adversarial vulnerability, underscoring the need to jointly design for selectivity and robustness in future attention mechanisms.

## 1 INTRODUCTION

Recent advances in attention mechanisms have significantly enhanced the representational capacity of deep learning models, driving breakthroughs across vision, language, and multimodal domains. However, these mechanisms can sometimes lead to contextual hallucinations due to the misallocation of attention scores (Huang et al., 2024; Maynez et al., 2020).

The Differential Transformer (Ye et al., 2025), introduced alongside the Differential Attention (DA) mechanism, offers a promising refinement to address such misallocation issues. DA employs two distinct attention maps, $A_1$ and $A_2$, and introduces a subtraction operation, $A_1 - \lambda A_2$, inspired by noise-cancellation techniques in signal processing. This formulation suppresses irrelevant or noisy information while enhancing focus on task-relevant features. The subtraction structure encourages higher attention weights in $A_1$ for informative regions and lower weights in $A_2$ for the same regions, yielding sharper and more selective outputs (Figure 1(a)).

Thanks to these advantages, the Differential Transformer effectively mitigates contextual hallucinations in summarization and question answering tasks (Ye et al., 2025). This improvement likely stems from DA's increased selectivity for task-relevant content, aligning with prior findings (Huang et al., 2024) that identify attention misallocation as a key cause of hallucination in standard Transformers.

These properties make DA particularly attractive for safety-critical applications, such as autonomous driving, medical diagnostics, and legal document analysis, where minimizing spurious or incoherent model behavior is essential. These advantages have already inspired a number of follow-up studies (Abid et al., 2025; Hammoud & Ghanem, 2025; Schneider et al., 2025).

---

[*]Currently with Acompany Co., Ltd. [†]Work done while the author was an intern at Turing Inc.

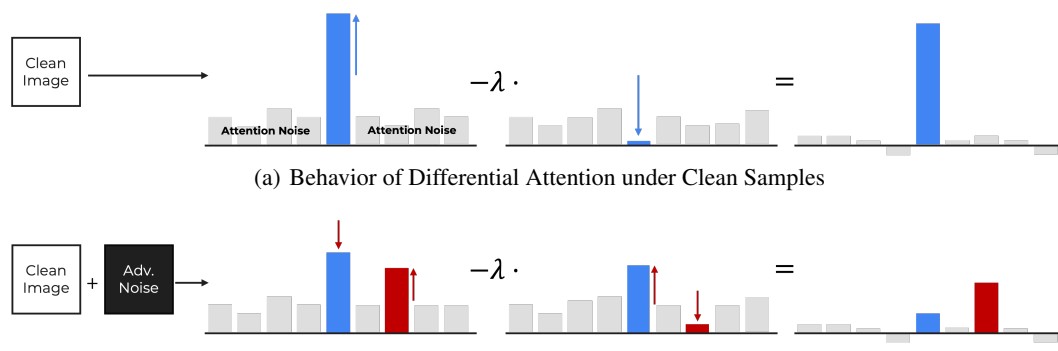

(a) Behavior of Differential Attention under Clean Samples

(b) Behavior of Differential Attention under Adversarial Samples

Figure 1: Illustration of the Fragile Principle of Differential Attention. (a) On clean inputs, well-aligned attention maps cancel redundant focus, producing sharp and stable responses. (b) With adversarial perturbations, the gradients (red arrows) of the two attention branches may become negatively aligned, amplifying small input changes and leading to conflicting responses.

At first glance, the subtractive structure of DA may appear inherently beneficial for robustness—by attenuating noisy signals, one might expect it to improve stability against adversarial perturbations as well. However, does this intuition hold? In this work, we rigorously challenge this assumption and show that the same structure designed to suppress noise can introduce a latent vulnerability. We formalize this phenomenon as the *Fragile Principle* of DA, revealing how subtractive mechanisms amplify input perturbation sensitivity and expose the model to adversarial fragility (Figure 1(b)).

We theoretically show that when the gradients of the two branches are aligned in opposite directions—that is, they point against each other with $\cos\theta < 0$, a condition we call **negative gradient alignment**—the subtraction amplifies the overall gradient norm and increases the local Lipschitz constant. This phenomenon, inherent to DA's subtractive design, not only reduces attention misallocation but also increases sensitivity to adversarial perturbations. Beyond this single-layer fragility, we uncover a **depth-dependent robustness effect**: stacking DA layers enhances robustness to small perturbations via cumulative noise cancellation, though fragility re-emerges at larger budgets.

To validate this, we conduct extensive experiments on DA-based models trained on diverse datasets, evaluating their robustness against adversarial perturbations. The results support our theoretical predictions: DA exhibits higher attack success rates, more frequent negative gradient alignment, and larger local Lipschitz estimates than models with standard attention.

**Contributions.** Our work makes the following key contributions:

- We present the first theoretical investigation of Differential Attention (DA) from the perspective of adversarial robustness.
- We show that DA's subtraction, while mitigating attention misallocation, structurally induces *negative gradient alignment*, amplifying sensitivity compared to standard attention and revealing a unique adversarial vulnerability (Section 4).
- We develop a depth-dependent analysis, theoretically and empirically demonstrating how cumulative noise cancellation yields partial robustness under small perturbations (Section 4).
- We evaluate DiffCLIP and DiffViT across multiple datasets, showing increased attack success rates, stronger negative gradient alignment, and higher local Lipschitz estimates relative to standard attention (Section 5).

## 2 RELATED WORK

**Attention Robustness and Vulnerability.** Transformers (Vaswani et al., 2017) achieve state-of-the-art results across domains, yet are more vulnerable to adversarial inputs than CNNs (Bai et al., 2021b; Mo et al., 2022; Jain & Dutta, 2024). ViTs (Dosovitskiy et al., 2021) are especially susceptible to patch attacks due to global receptive fields (Fu et al., 2022; Liu et al., 2023). Transfer-based

attacks (Naseer et al., 2022; Ming et al., 2024) and robustness-oriented training (Qin et al., 2022; Wang et al., 2021) show that robustness must be engineered. Differential Attention (DA) suppresses redundant signals through subtraction, but its robustness remains unexplored. While prior works aim to improve attention robustness through architectural modifications or robustness-enhancing training, our work takes a different perspective: we analyze the intrinsic sensitivity of DA as a fixed mechanism, rather than proposing or comparing robustness-oriented variants.

**Interpretability in Attention.** The explanatory role of attention is contested: some argue it fails to reflect reasoning (Jain & Wallace, 2019; Serrano & Smith, 2019; Bai et al., 2021a), while others promote sparsity or concept alignment (Martins et al., 2020; Pan et al., 2021; Rigotti et al., 2022). Structured mechanisms such as Structured Attention (Niculae & Blondel, 2017) aim for transparency, but their robustness implications remain unclear.

**Lipschitz Behavior and Gradient Sensitivity.** Certified robustness often relies on Lipschitz constraints, which bound worst-case changes. Prior work constrains attention layers (Kim et al., 2021; Dasoulas et al., 2021), stabilizes entropy (Zhai et al., 2023), or shows weight decay modulates sensitivity (Kobayashi et al., 2024). Other analyses examine sequence length and normalization (Castin et al., 2024). These works suggest Lipschitz properties strongly shape robustness.

**Structural Fragility and OOD Sensitivity.** Beyond input perturbations, structural vulnerabilities also emerge: CNNs exhibit frequency-specific sensitivity (Tsuzuku & Sato, 2019), and flows assign high likelihood to trivial OODs (Osada et al., 2024). These findings underscore that architecture itself can amplify or dampen robustness. To our knowledge, no prior work studies DA's subtractive design, which we show suppresses redundancy yet paradoxically increases adversarial fragility.

# 3 PRELIMINARY

In this section, we briefly review the standard attention mechanism and introduce the structure of Differential Attention (DA) (Ye et al., 2025), which forms the basis for our theoretical analysis.

## 3.1 ATTENTION MECHANISM

Given an input sequence represented as a matrix $X \in \mathbb{R}^{N \times d}$, where $N$ is the sequence length and $d$ is the embedding dimension, the self-attention mechanism computes the output as follows:

$$\text{Attention}(X) = \text{Softmax}\left(\frac{QK^\top}{\sqrt{d_k}}\right)V,$$

where the query, key, and value matrices are linear projections: $Q = XW^Q, K = XW^K, V = XW^V$, with $W^Q, W^K, W^V \in \mathbb{R}^{d \times d_k}$ being learned parameters, and $d_k$ denoting the head dimension. The softmax operation normalizes the attention scores across the key dimension, assigning higher weights to tokens that are more relevant to each query.

## 3.2 DIFFERENTIAL ATTENTION

Differential Attention (DA) (Ye et al., 2025) modifies the standard attention structure by introducing two parallel branches of attention maps, whose outputs are combined in a subtractive manner to suppress redundant or noisy signals. Specifically, DA computes two separate sets of query and key projections:

$$Q_1 = XW_1^Q, \quad K_1 = XW_1^K, \quad Q_2 = XW_2^Q, \quad K_2 = XW_2^K,$$

and a single shared value projection $V = XW^V$ as well as standard attention. Each branch independently computes its own attention map as

$$A_1 = \text{Softmax}\left(\frac{Q_1K_1^\top}{\sqrt{d_k}}\right), \quad A_2 = \text{Softmax}\left(\frac{Q_2K_2^\top}{\sqrt{d_k}}\right).$$

The output of DA is then given by:

$$\text{DA}(X) = (A_1 - \lambda A_2)V,$$

where $\lambda$ is a learned or fixed scalar parameter that controls the contribution of the subtractive branch. When $\lambda$ is trainable, its initial value is set by $\lambda_{\text{init}}$. The default value of $\lambda_{\text{init}}$ is set to 0.8 through intensive study in (Ye et al., 2025). This structure aims to enhance semantic focus by amplifying meaningful patterns captured by $A_1$ while suppressing redundant or noisy patterns captured by $A_2$.

## 4 THEORETICAL ANALYSIS: STRUCTURAL SENSITIVITY AND ROBUSTNESS

We now provide a theoretical analysis of the sensitivity of Differential Attention (DA) to input perturbations. While DA is motivated by the goal of suppressing redundant focus through subtraction, we show that the same structure can, under specific conditions, amplify sensitivity and thereby introduce fragility.

We first examine how gradient amplification arises when the two attention branches are negatively aligned—a configuration that DA naturally encourages (Section 4.1). We then extend the analysis to multiple layers, where stacking can yield a depth-dependent cancellation effect (Section 4.2). Finally, we discuss the scope and limitations of these results.

**Notation.** Let $x$ be a clean input and $x' = x + \xi$ a perturbed input with small perturbation $\xi$. We analyze the sensitivity of the differential attention map $A_{\text{DA}}(x)$ with respect to $\xi$. We denote by $A_{\text{base}}$ the standard (non-differential) attention map.

All proofs of Lemmas and Theorems are provided in Appendix A.

### 4.1 FRAGILE PRINCIPLE: SENSITIVITY AMPLIFICATION VIA THE SUBTRACTIVE STRUCTURE

A core feature of DA is its subtractive formulation: $A_{\text{DA}} = A_1 - \lambda A_2$, designed to suppress redundant or noisy attention patterns. For this subtraction to be effective, the two attention maps $A_1$ and $A_2$ cannot act independently; they must focus on overlapping regions but with opposing intensities (see Figure 1). In practice, this means that during training the model is implicitly encouraged to assign gradients pointing in opposite directions to $A_1$ and $A_2$ within those regions.

Such opposite alignment is not merely a byproduct but a functional necessity: without it, the subtraction would fail to sharpen focus or remove noise. The cancellation mechanism therefore builds in a structural bias toward negative gradient alignment. While this property enables DA to highlight informative regions more clearly than standard attention, it also raises the possibility that the same mechanism could amplify sensitivity to perturbations. Our analysis begins from this structural observation and investigates its consequences for robustness.

**Lemma 1.** Let $\theta$ be the angle between the input gradients of $A_1$ and $A_2$. Then,

$$\|\nabla_\xi A_{\text{DA}}\|^2 = \|\nabla_\xi A_1\|^2 + \lambda^2 \|\nabla_\xi A_2\|^2 - 2\lambda \|\nabla_\xi A_1\| \|\nabla_\xi A_2\| \cos\theta. \tag{1}$$

The cross-term becomes positive whenever $\cos\theta < 0$, causing gradient amplification.

**Theorem 1** (Sensitivity Amplification by Alignment). Let $\rho = \|\nabla_\xi A_2\| / \|\nabla_\xi A_1\|$. Then,

$$\|\nabla_\xi A_{\text{DA}}\|^2 = \|\nabla_\xi A_1\|^2 \left(1 + \lambda^2 \rho^2 - 2\lambda\rho \cos\theta\right).$$

Asymptotically,

$$\|\nabla_\xi A_{\text{DA}}\| = \begin{cases} (1 - \lambda\rho) \|\nabla_\xi A_1\| & \text{if } \cos\theta = +1, \\ (1 + \lambda\rho) \|\nabla_\xi A_1\| & \text{if } \cos\theta = -1. \end{cases} \tag{2}$$

Thus, under negative alignment ($\cos\theta < 0$), DA inherently amplifies perturbation sensitivity. This fragility arises not by accident but as a structural byproduct of DA's cancellation goal.

We now compare DA's sensitivity to that of standard attention maps.

**Theorem 2** (Relative Sensitivity to Standard Attention). Let $A_{\text{DA}} = A_1 - \lambda A_2$ and $A_{\text{base}}$ be standard attention. Define $\gamma = \frac{\|\nabla_\xi A_1\|}{\|\nabla_\xi A_{\text{base}}\|}$. Then,

$$\frac{\|\nabla_\xi A_{\text{DA}}\|}{\|\nabla_\xi A_{\text{base}}\|} = \gamma \sqrt{1 + \lambda^2 \rho^2 - 2\lambda\rho \cos\theta}, \tag{3}$$

with maximum attained when $\cos\theta = -1$ and minimum when $\cos\theta = +1$.

**Theorem 3** (Existence of Amplifying Perturbations). *There exists a perturbation $\xi$ such that*

$$\frac{\|\nabla_\xi A_{\mathrm{DA}}\|}{\|\nabla_\xi A_{\mathrm{base}}\|} > 1 \quad \text{if and only if} \quad \cos\theta < \frac{1 + \lambda^2\rho^2 - \gamma^{-2}}{2\lambda\rho}.$$

This condition delineates the $(\rho, \theta)$ region where DA exhibits strictly higher sensitivity than standard attention. Since $\rho$ and $\theta$ can be adversarially controlled, DA's subtractive design exposes a structural vulnerability.

**Implication for Lipschitz constants.** To assess how the above gradient amplification affects robustness, we consider the *local Lipschitz constant*, which captures the worst-case sensitivity of the attention map to small perturbations. Relating DA's gradient behavior to Lipschitz continuity reveals that subtraction structurally shifts sensitivity upward, often increasing local Lipschitz values and weakening robustness.

Building on the definition of the local Lipschitz constant,

$$L(x) = \sup_{\xi \neq 0} \frac{\|A(x + \xi) - A(x)\|_2}{\|\xi\|}, \tag{4}$$

we derive the following bound:

**Lemma 2.** *Let $L_{\mathrm{DA}}(x)$ and $L_{\mathrm{base}}(x)$ be the local Lipschitz constants of $A_{\mathrm{DA}}$ and $A_{\mathrm{base}}$. Then,*

$$\frac{L_{\mathrm{DA}}(x)}{L_{\mathrm{base}}(x)} \leq \gamma\sqrt{1 + \lambda^2\rho^2 - 2\lambda\rho\cos\theta}.$$

This inequality shows explicitly how the subtraction weight $\lambda$ and the gradient alignment angle $\theta$ together govern DA's local sensitivity. We further analyze the certifiable robustness radius in Appendix B, extending this discussion.

## 4.2 DEPTH-DEPENDENT ROBUSTNESS VIA NOISE CANCELLATION

**Noise cancellation as a structural property of DA.** DA is designed with a subtractive form, $A_{\mathrm{DA}} = A_1 - \lambda A_2$, which suppresses components common to both $A_1$ and $A_2$. This *noise cancellation effect* is independent of gradient alignment: it does not rely on whether $\nabla A_1$ and $\nabla A_2$ are aligned or anti-aligned, but rather on the structural subtraction that systematically reduces shared activations or perturbations. The strength of this effect is governed primarily by the coefficient $\lambda$.

**Accumulation across depth.** When DA layers are stacked, this cancellation compounds: shared noise that survives one layer is more likely to be attenuated again in subsequent layers. In contrast, standard attention lacks such a subtractive mechanism, so its effective perturbation propagation is closer to a simple accumulation across depth. Formally, if $f^{(d)}$ denotes the $d$-th DA layer with local Lipschitz constant $L_{\mathrm{DA}}^{(d)}$, and $\alpha^{(d)} \in (0, 1]$ denotes the average reduction factor from noise cancellation at layer $d$, then

$$\|\Delta^{(d)}\| \leq \alpha^{(d)} L_{\mathrm{DA}}^{(d)} \|\Delta^{(d-1)}\|, \qquad \Delta^{(0)} = \xi.$$

**Consequence (upper bound).** By recursion, let $F = f^{(D)} \circ \cdots \circ f^{(1)}$ denote the $D$-layer composition. Then,

$$\|F(x + \xi) - F(x)\| \leq \left(\prod_{d=1}^{D} \alpha^{(d)} L_{\mathrm{DA}}^{(d)}\right) \|\xi\| = (\bar{\alpha}\,\bar{L}_{\mathrm{DA}})^D \|\xi\|, \tag{5}$$

where $\bar{\alpha}$ and $\bar{L}_{\mathrm{DA}}$ denote the geometric means across layers. If $\bar{\alpha}\bar{L}_{\mathrm{DA}} < 1$, small perturbations are attenuated with depth, yielding robustness gains that are *specific to DA*.

We note that $\alpha$ is not an independent hyperparameter but rather an emergent factor that is influenced by $\lambda$ as well as the distribution of activations across layers. Intuitively, larger $\lambda$ magnifies the subtractive effect, making it statistically more likely that perturbations are partially cancelled across layers, and thus biases $\alpha$ downward. However, the exact value of $\alpha$ is also shaped by training dynamics, and cannot be reduced to a deterministic function of $\lambda$.

**Theorem 4** (Depth-Dependent Sensitivity of Standard Attention vs. DA). For standard attention, perturbations propagate as

$$\|\Delta^{(D)}\| \le (\bar{L}_{\text{base}})^D \|\xi\|,$$

with no cancellation effect ($\alpha^{(d)} \approx 1$), unlike DA's structural differentiation. For DA, the bound becomes

$$\|\Delta^{(D)}\| \le (\bar{\alpha}\,\bar{L}_{\text{DA}})^D \|\xi\|,$$

where $\bar{\alpha} < 1$ reflects structural noise cancellation.

**Corollary 1** (Crossover in Robustness). If $\bar{L}_{\text{DA}} > \bar{L}_{\text{base}}$ (DA is locally more sensitive) but $\bar{\alpha} < 1$ (nontrivial cancellation), then there exists a depth threshold $D^*$ such that

$$(\bar{\alpha}\,\bar{L}_{\text{DA}})^{D^*} = (\bar{L}_{\text{base}})^{D^*}.$$

For $D < D^*$, DA is more fragile than standard attention; for $D > D^*$, DA becomes asymptotically more robust.

**Interpretation.** This analysis highlights two independent mechanisms in DA: (i) negative gradient alignment, which locally amplifies fragility, and (ii) noise cancellation, which systematically attenuates shared perturbations and strengthens with depth. The coexistence of these forces explains why DA can appear fragile at shallow depths yet exhibit improved robustness when scaled deeper. In particular, perturbations amplified in early layers do not propagate unchanged: each subsequent DA layer introduces its own cancellation step, so the perturbation energy is progressively reduced as depth increases. This cumulative attenuation provides a simple explanation for the empirical trend we observe—strong fragility in shallow models and partial robustness in deeper stacks—without requiring additional architectural assumptions.

### 4.3 LIMITATIONS OF THE THEORETICAL ANALYSIS

While our analysis offers a structural understanding of the sensitivity amplification in DA, it is based on a few simplifying assumptions that may not always hold in practice:

**Local linear approximation via input gradients.** Our derivation relies on local gradient-based sensitivity—i.e., assuming that local behavior of $A_{\text{DA}}$ can be well-approximated by its gradient. This is valid under small perturbations, but may not fully capture global nonlinear effects in deep networks. This local first-order assumption follows the standard practice in prior analyses of attention robustness (e.g., Kim et al. (2021); Dasoulas et al. (2021)), which similarly characterize sensitivity through gradients under small perturbations. This regime matches typical adversarial settings (e.g., PGD or CW with $\epsilon \le 8/255$), where first-order effects dominate the model's response. Our theoretical results should therefore be interpreted as local sensitivity characterizations rather than global guarantees.

**Layer isolation.** We analyze DA in isolation, holding other network layers fixed. In practice, interactions with downstream modules may either mitigate or exacerbate sensitivity.

These assumptions are common in robustness analyses. They clarify how DA's design induces fragility, but also underscore the need for empirical validation (Section 5).

## 5 EXPERIMENTS

We conduct extensive experiments to validate our theoretical analysis of the Fragile Principle in DA. Specifically, we evaluate: (i) adversarial vulnerability measured by attack success rate (ASR), (ii) the frequency of negative gradient alignment ($\cos(\nabla_\xi A_1, \nabla_\xi A_2) < 0$), and (iii) empirical estimates of the local Lipschitz constant. We also examine how these effects vary with model depth.

**Scope of our evaluation.** Our experiments are designed to isolate the architectural effect of DA's subtractive structure. For this reason, we train and evaluate each model on the same dataset and do not incorporate robustness-enhancing training schemes, which would change many other aspects of the representation and thus confound the analysis of DA itself. Similarly, we focus on well-established and broadly used attack baselines (PGD (Madry et al., 2018), AutoAttack (Croce & Hein, 2020b), Carlini–Wagner (CW) (Carlini & Wagner, 2017)) rather than task- or model-specific adversarial methods, since the latter may introduce additional factors unrelated to the mechanism we study.

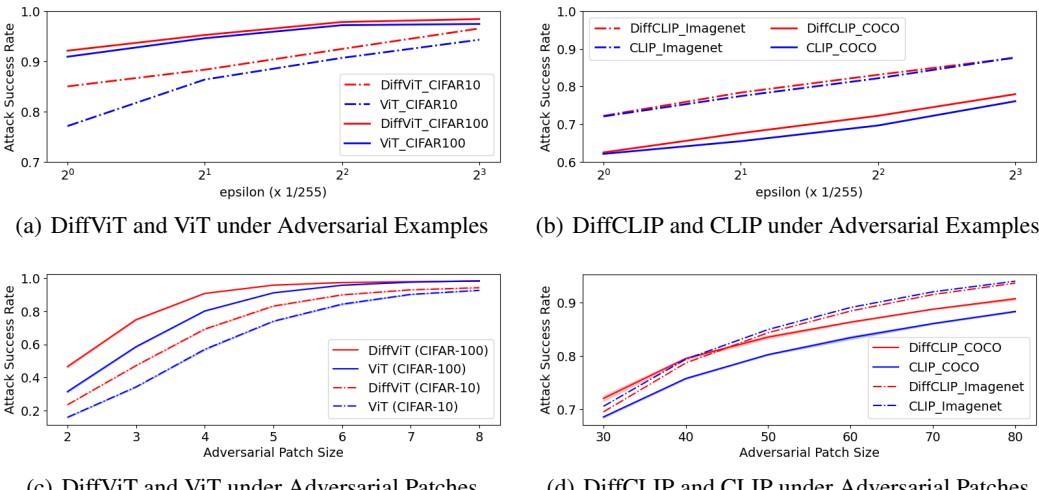

(a) DiffViT and ViT under Adversarial Examples      (b) DiffCLIP and CLIP under Adversarial Examples

(c) DiffViT and ViT under Adversarial Patches      (d) DiffCLIP and CLIP under Adversarial Patches

Figure 2: ASR under Adversarial Examples and Patches crafted with PGD. DiffViT and DiffCLIP generally exhibit higher or comparable ASR compared to standard attention, with the gap most pronounced in small-class datasets (CIFAR and COCO), while narrowing on large-scale ones (Imagenet).

## 5.1 EXPERIMENTAL SETUP

**Models.** We evaluate two classes of models: (1) ViT and DiffViT trained from for controlled studies, and (2) pre-trained CLIP (ViT-B/16) (Radford et al., 2021) and its DA-based variant DiffCLIP (Hammoud & Ghanem, 2025). The ViT/DiffViT models are lightweight transformers with a $D$-layer Attention or Differential Attention (DA) module, where $D \in \{1, 2, 4, 8, 12\}$. Unless otherwise stated, we use $D = 1$. The subtraction coefficient $\lambda$ is initialized at $\lambda_{\text{init}} = 0.8$ and updated during training, following (Ye et al., 2025). DiffCLIP extends CLIP by replacing standard attention with DA, while preserving CLIP's contrastive training objective between image and text. This substitution allows DiffCLIP to suppress redundant activations and sharpen focus on discriminative regions. In addition, (Hammoud & Ghanem, 2025) introduce DiffViT, a ViT variant that employs DA.

**Datasets.** For ViT and DiffViT, we train on CIFAR-10 and CIFAR-100 (Krizhevsky, 2009), consisting of 10 and 100 classes of $32 \times 32$ natural images. We also include Tiny ImageNet (Le & Yang, 2015) for ViT and DiffViT with depth 4, with results reported in Appendix. For CLIP and DiffCLIP, we use the MSCOCO 2014 validation set (Lin et al., 2014) (40k images annotated by 80 categories) and the ImageNet-1k validation set (Deng et al., 2009) (50k images across 1,000 categories).

**Adversarial Attacks.** We consider PGD (Madry et al., 2018), Carlini–Wagner (CW) (Carlini & Wagner, 2017), and AutoAttack (AA) (Croce & Hein, 2020b). PGD is applied both as an adversarial example attack and as an adversarial patch attack. For CLIP and DiffCLIP, perturbations minimize the cosine similarity between the perturbed image encoding $\phi_{\text{visual}}(x + \delta)$ and its text prompt embedding $\phi_{\text{text}}(t)$, where $t$ is a class description (e.g., "a photo of <class>"). For ViT and DiffViT, perturbations instead minimize cross-entropy loss to induce misclassification. PGD and AutoAttack are constrained under $\ell_\infty$ norm, while CW is applied under $\ell_2$. Patch attacks restrict $\delta$ to a square of size $w \times w$ at a random image location.

All experiments are run on a single NVIDIA H100 GPU. Implementation details and hyperparameters are provided in Appendix C.

## 5.2 ATTACK SUCCESS RATE

To assess adversarial vulnerability, we compute the attack success rate (ASR):

$$\text{ASR} = \frac{\#\text{successful attacks}}{\#\text{trials}}.$$

Since we focus on untargeted attacks, a perturbation is successful if it changes the prediction relative to the clean input.

Table 1: Attack success rates and clean accuracy of DiffViT on CIFAR-10 when varying $\lambda_{\text{init}}$. ASR increases with $\lambda_{\text{init}}$ up to 0.8. Beyond that point, ASR drops, suggesting over-cancellation.

| $\lambda_{\text{init}}$ | 0.5 | 0.7 | 0.8 (default) | 0.85 | 0.9 | 0.95 |
|---|---|---|---|---|---|---|
| Accuracy | 0.8605 | 0.8697 | **0.8700** | 0.8567 | 0.8524 | 0.8468 |
| ASR ($\epsilon = 1/255$) | 0.4074 | 0.6772 | **0.8498** | 0.7531 | 0.4956 | 0.4164 |

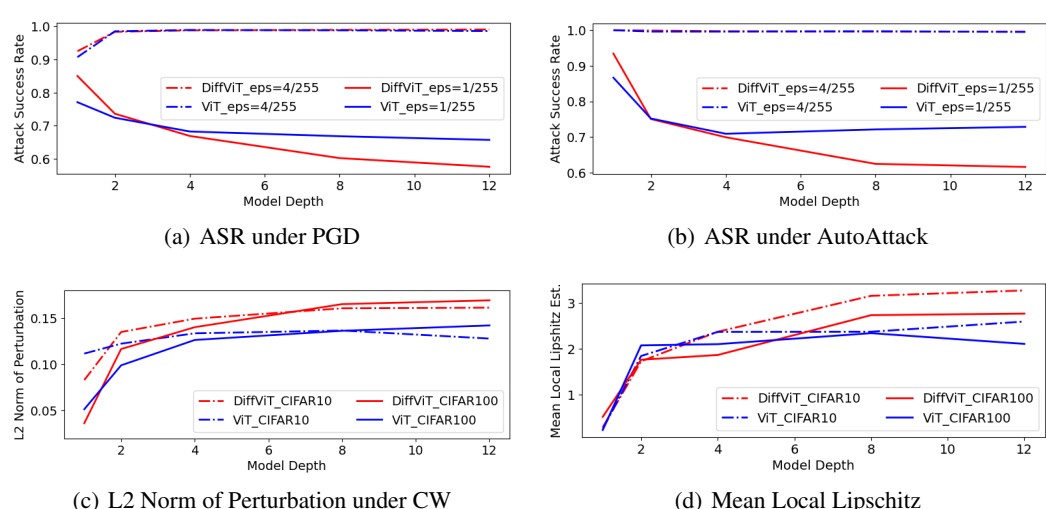

(a) ASR under PGD

(b) ASR under AutoAttack

(c) L2 Norm of Perturbation under CW

(d) Mean Local Lipschitz

Figure 3: Depth-dependent effects of DA. (a) (b) Under PGD and AutoAttack, DA is fragile at depth 1–2, but ASR drops with depth for $\epsilon=1/255$; at $\epsilon=4/255$ both converge to high ASR. (c) Under CW-L2, deeper models require larger perturbations to reach 100% ASR. (d) Mean local Lipschitz estimates over all layers rise with depth, indicating higher local sensitivity.

**Controlled Study on ViT/DiffViT (Single Layer).** Figure 2(a) and 2(c) show ASR under adversarial examples and patch attacks, respectively. Across all cases, DiffViT exhibits higher ASR than ViT, consistent with the Fragile Principle. We also vary $\lambda_{\text{init}}$, which controls the subtraction strength in DA. Table 1 shows ASR on CIFAR-10 increases steadily up to $\lambda_{\text{init}} = 0.8$, after which it declines, suggesting over-subtraction reduces vulnerability.

**Controlled Study on ViT/DiffViT (Multiple Layers).** Our theory predicts that although a single DA layer embodies the Fragile Principle by amplifying sensitivity, stacking layers reverses this trend through a cancellation effect, producing depth-dependent robustness. Figures 3(c) support this: under CW, deeper DiffViTs require larger perturbations to reach ASR=100% (Fig. 3(c)). Under PGD and AutoAttack, those ASRs are highest at depth 1 but decreases with depth for $\epsilon=1/255$, surpassing ViT's robustness (Fig. 3(a)); at $\epsilon=4/255$, both saturate at high ASR. We report additional experiments on Tiny ImageNet in Appendix D.1, which show the same trend.

**Pretrained Models (CLIP/DiffCLIP).** Figure 2(b) and 2(d) report ASR for CLIP and DiffCLIP evaluated on COCO and ImageNet. On COCO, DiffCLIP consistently shows higher vulnerability across perturbation budgets and patch sizes, supporting the Fragile Principle. On ImageNet, results are more nuanced: under adversarial examples, DiffCLIP remains slightly more vulnerable, whereas under patch attacks CLIP marginally outperforms DiffCLIP, though the difference is small. This suggests that dataset scale and diversity can modulate the manifestation of DA's fragility.

## 5.3 SENSITIVITY AMPLIFICATION UNDER PERTURBATIONS

We next test whether DA amplifies local sensitivity. We estimate the local Lipschitz constant (4), sampling $\xi$ uniformly from $\ell_\infty$-bounded noise with $\|\xi\| \leq 8/255$.

Figure 3(d) reports the *mean local Lipschitz estimate*, computed by averaging the local Lipschitz constants across all layers of each model. We observe that deeper DiffViT models consistently yield

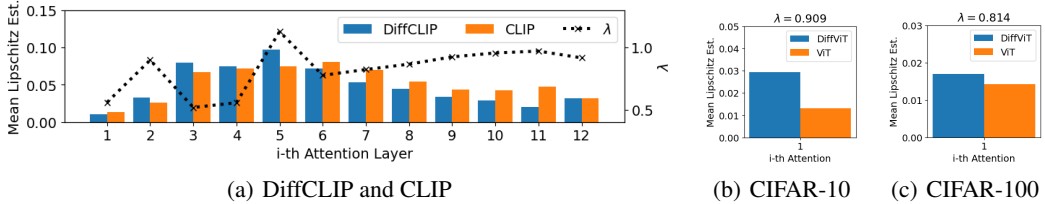

Figure 4: Mean Lipschitz estimates of attention layers under input perturbations. Models incorporating DA exhibit the highest values among all attention layers, particularly at layers with larger $\lambda$.

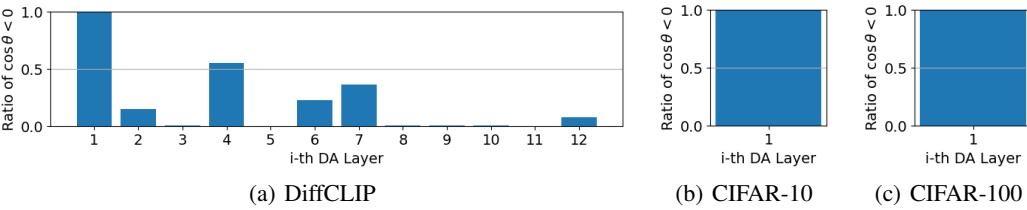

Figure 5: Frequency of negative gradient alignment $(\cos(\nabla A_1, \nabla A_2) < 0)$ at each layer.

higher per-layer Lipschitz values, indicating that local sensitivity increases with depth. Figure 4 further compares ViT/DiffViT and CLIP/DiffCLIP. In all settings, DA-equipped models exhibit significantly higher Lipschitz estimates than baselines, with maxima at layers using larger $\lambda$. Interestingly, later layers often display lower sensitivity than earlier ones, suggesting that the cumulative dynamics of DA distribute perturbation effects unevenly across depth rather than simply compounding them. The wider spread of values in DA models highlights their more variable sensitivity across layers.

These results confirm that DA's subtractive design can both amplify and reduce sensitivity, aligning with our theoretical claims. The amplified sensitivity facilitates adversarial perturbations, underscoring risks in safety-critical domains.

### 5.4 NEGATIVE GRADIENT ALIGNMENT IN DIFFERENTIAL ATTENTION

Finally, we examine whether DA structurally induces negative gradient alignment. For perturbations $\xi$ bounded by $\|\xi\| \leq 8/255$, we compute cosine similarity between gradients $\nabla_\xi A_1$ and $\nabla_\xi A_2$, and record the frequency of $\cos < 0$.

Figure 5 summarizes results. In DiffCLIP, negative alignment is strongest in the first layer but still present deeper. For CIFAR-trained ViT/DiffViT, even the simplest single-layer variants show dominant negative alignment.

Thus, negative gradient alignment is not a rare anomaly but a structural property of DA, providing direct empirical evidence for the Fragile Principle.

## 6 DISCUSSION AND LIMITATION

**Summary of findings.** Our study reveals a structural trade-off at the heart of Differential Attention (DA). While its subtractive design sharpens task-relevant focus and reduces contextual hallucination, it also inherently amplifies sensitivity when the two branches exhibit negative gradient alignment. This *Fragile Principle* manifests as increased local Lipschitz constants, higher attack success rates, and frequent opposing gradient flows. At the same time, our depth-dependent analysis shows that DA models can gain robustness at small perturbation budgets when stacking layers, owing to the cumulative effect of noise cancellation across depth. However, this protection diminishes as perturbation strength grows.

**Limitations.** Our theoretical analysis relies on local linearization and isolates the attention mechanism from other layers. These assumptions clarify the structural role of subtraction but may not capture

full nonlinear interactions. Moreover, our exploration of the subtraction weight $\lambda$ was limited to initialization; a fuller study of its training dynamics remains open. Another limitation lies in the controlled nature of our experiments. Since our aim is to isolate the mechanism-specific behavior of DA, we focus on small, gradient-based perturbations and do not evaluate robustness under natural or semantic adversarial examples (e.g., distribution shifts, physical noise, or natural-adversarial image curation). These factors introduce complex, task- and dataset-dependent effects that can obscure the structural phenomena we analyze. Extending DA's sensitivity analysis to such real-world perturbations is an important direction for future work, but is deliberately outside the scope of our mechanism-level study.

**Future directions for robustness.** Several avenues could mitigate DA's fragility. First, careful tuning of $\lambda$ offers a trade-off between noise suppression and adversarial robustness, as suggested in Table 1. Second, our results suggest that increasing DA depth itself can serve as a lightweight mitigation: deeper models required larger perturbations under CW and exhibited lower ASR under PGD/AA at small budgets, though this advantage disappears under stronger attacks (Fig. 3). Third, adversarial training with small perturbations reduced ASR (Appendix D.2), showing compatibility with standard defenses. Finally, we view DA as a natural candidate for integration with certified defenses, where its selective subtraction could complement Lipschitz-based guarantees.

Overall, this work highlights both the promise and the fragility of subtractive attention mechanisms, and we hope it inspires future designs that preserve DA's discriminative focus while mitigating its structural vulnerability.

## 7    CONCLUSION

We analyzed Differential Attention (DA) and revealed a structural fragility that arises under adversarial perturbations. While DA sharpens focus and suppresses contextual hallucination on clean inputs, its subtractive design can amplify local sensitivity via negative gradient alignment. This fragility reflects a fundamental trade-off: DA's discriminative benefits may come at the cost of adversarial vulnerability. We validated this trade-off through gradient alignment analysis, local Lipschitz estimation, and depth-dependent experiments. Importantly, our results show that increasing DA depth can naturally mitigate small perturbations via cumulative noise cancellation, although this effect diminishes under stronger attacks. These insights highlight the need to jointly consider clean performance, adversarial robustness, and architectural depth when designing future attention mechanisms.

**Ethics Statement.**    This work focuses on the theoretical and empirical analysis of Differential Attention (DA) under adversarial perturbations. Our study does not involve human subjects, personal or sensitive data, or any identifiable information. All datasets used in this paper are publicly available benchmarks (CIFAR-10/100, Tiny ImageNet, MSCOCO, ImageNet-1k) that are widely adopted in the research community and collected under ethical and legal guidelines. While our analysis reveals structural vulnerabilities in DA, the goal is to advance understanding of robustness in attention mechanisms and inspire more resilient designs. We do not foresee direct harmful applications of these findings; rather, highlighting DA's fragility is a necessary step toward safer deployment in safety-critical domains such as autonomous driving or medical AI.

**Reproducibility Statement.**    We have taken care to ensure the reproducibility of our results. All model architectures, training setups, and hyperparameters are described in the main text and Appendix C. For theoretical contributions, assumptions are explicitly stated, and proofs of lemmas and theorems are provided in Appendix A. For empirical evaluation, we detail dataset specifications (CIFAR-10/100, Tiny ImageNet, MSCOCO, ImageNet-1k) and attack configurations (PGD, CW, AutoAttack), including perturbation budgets and norm constraints. Results are consistently reported across multiple datasets and model configurations (e.g., depth, $\lambda$ initialization) to support robustness of conclusions. Figures and tables in the main text and Appendix provide complete experimental outcomes, ensuring transparency and replicability.

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

## A  PROOF OF THEORETICAL CLAIMS

This section provides complete proofs of the lemmas and theorems presented in Section 4.

### A.1  PROOF OF LEMMA 1

*Proof.* Let $\theta$ be the angle between the input gradients of $A_1$ and $A_2$, i.e., $\cos\theta = \frac{\langle\nabla_\xi A_1, \nabla_\xi A_2\rangle}{\|\nabla_\xi A_1\|\cdot\|\nabla_\xi A_2\|}$. From the definition of DA, $A_{\mathrm{DA}} = A_1 - \lambda A_2$, we compute its gradient with respect to input perturbation $\xi$: $\nabla_\xi A_{\mathrm{DA}} = \nabla_\xi A_1 - \lambda\nabla_\xi A_2$. Taking the squared norm:

$$\|\nabla_\xi A_{\mathrm{DA}}\|^2 = \|\nabla_\xi A_1\|^2 + \lambda^2\|\nabla_\xi A_2\|^2 - 2\lambda\langle\nabla_\xi A_1, \nabla_\xi A_2\rangle$$
$$= \|\nabla_\xi A_1\|^2 + \lambda^2\|\nabla_\xi A_2\|^2 - 2\lambda\|\nabla_\xi A_1\|\|\nabla_\xi A_2\|\cos\theta.$$

$\square$

### A.2  PROOF OF THEOREM 1

*Proof.* By substituting $\rho = |\nabla_\xi A_2|/|\nabla_\xi A_1|$ into the result of Lemma 1, we obtain:

$$\|\nabla_\xi A_{\mathrm{DA}}\|^2 = \|\nabla_\xi A_1\|^2\left(1 + \lambda^2\rho^2 - 2\lambda\rho\cos\theta\right).$$

**Case 1:** If $\cos\theta = +1$ (aligned gradients), then:

$$\|\nabla_\xi A_{\mathrm{DA}}\|^2 = \|\nabla_\xi A_1\|^2\left(1 + \lambda^2\rho^2 - 2\lambda\rho\right) = \|\nabla_\xi A_1\|^2(1 - \lambda\rho)^2.$$

**Case 2:** If $\cos\theta = -1$ (oppositely aligned gradients), then:

$$\|\nabla_\xi A_{\mathrm{DA}}\|^2 = \|\nabla_\xi A_1\|^2\left(1 + \lambda^2\rho^2 + 2\lambda\rho\right) = \|\nabla_\xi A_1\|^2(1 + \lambda\rho)^2.$$

$\square$

### A.3  PROOF OF THEOREM 2

*Proof.* Now define $\gamma := \|\nabla_\xi A_1\|/\|\nabla_\xi A_{\mathrm{base}}\|$. Substituting this into the expression derived in Theorem 1, we get:

$$\|\nabla_\xi A_{\mathrm{DA}}\|^2 = \gamma^2\|\nabla_\xi A_{\mathrm{base}}\|^2(1 + \lambda^2\rho^2 - 2\lambda\rho\cos\theta).$$

Taking square roots on both sides gives the desired result (3). Finally, note that this expression is minimized when $\cos\theta = +1$ (fully aligned), and maximized when $\cos\theta = -1$ (fully opposed), as the cross term $-2\lambda\rho\cos\theta$ becomes most negative or positive, respectively. $\square$

### A.4  PROOF OF THEOREM 3

*Proof.* Let us consider the set of perturbations $\xi$ with $\|\xi\| \le \epsilon$. Since the gradients $\nabla_\xi A_1$ and $\nabla_\xi A_2$ vary continuously with direction, there exists a direction of $\xi$ within this $\epsilon$-ball such that the angle $\theta$ satisfies the required condition. Thus, if the inequality $\cos\theta < \frac{1 + \lambda^2\rho^2 - \gamma^{-2}}{2\lambda\rho}$ is satisfied for some value of $\theta$ (i.e., some $\xi$), then such a direction $\xi$ with $|\xi| \le \epsilon$ exists, as the set of possible angles $\theta$ spans a continuous range over this ball. Since $\theta$ varies continuously with $\xi$, and $\cos\theta$ spans a continuous range, the intermediate value theorem ensures the existence of such a $\xi$. $\square$

### A.5  PROOF OF LEMMA 2

*Proof.* Recall that the local Lipschitz constant of a function $f$ at input $x$ is defined as: $L_f(x) = \sup_{\|\xi\|\neq 0}\frac{\|f(x+\xi) - f(x)\|}{\|\xi\|}$. If $f$ is differentiable at $x$, then: $L_f(x) \le \|\nabla_x f(x)\|$. We apply this to the attention maps. Since $A_{\mathrm{DA}}$ is differentiable almost everywhere, we have: $L_{\mathrm{DA}}(x) \le \|\nabla_\xi A_{\mathrm{DA}}(x)\|$. Similarly, $L_{\mathrm{base}}(x) \le \|\nabla_\xi A_{\mathrm{base}}(x)\|$.

Now, from Theorem 2, we already have the following relationship: $\frac{\|\nabla_\xi A_{\mathrm{DA}}(x)\|}{\|\nabla_\xi A_{\mathrm{base}}(x)\|} = \gamma\sqrt{1 + \lambda^2\rho^2 - 2\lambda\rho\cos\theta}$. Using the fact that the Lipschitz constant is upper-bounded by the gradient norm: $\frac{L_{\mathrm{DA}}(x)}{L_{\mathrm{base}}(x)} \le \frac{\|\nabla_\xi A_{\mathrm{DA}}(x)\|}{\|\nabla_\xi A_{\mathrm{base}}(x)\|} = \gamma\sqrt{1 + \lambda^2\rho^2 - 2\lambda\rho\cos\theta}$. $\square$

A.6 PROOF OF THEOREM 4

For standard attention, the layerwise deviation satisfies

$$\|\Delta^{(d)}\| \leq L_{\text{base}}^{(d)}\|\Delta^{(d-1)}\|,$$

which unrolls to

$$\|\Delta^{(D)}\| \leq \Big( \prod_{d=1}^{D} L_{\text{base}}^{(d)} \Big)\|\xi\| = (\bar{L}_{\text{base}})^D\|\xi\|.$$

For DA, the subtractive form introduces an additional contraction factor $\alpha^{(d)} \in (0, 1]$ per layer, giving

$$\|\Delta^{(d)}\| \leq \alpha^{(d)} L_{\text{DA}}^{(d)}\|\Delta^{(d-1)}\|.$$

Thus

$$\|\Delta^{(D)}\| \leq \Big( \prod_{d=1}^{D} \alpha^{(d)} L_{\text{DA}}^{(d)} \Big)\|\xi\| = (\bar{\alpha}\,\bar{L}_{\text{DA}})^D\|\xi\|,$$

where $\bar{\alpha}$ and $\bar{L}_{\text{DA}}$ denote geometric means. Since DA's subtractive structure systematically biases $\alpha^{(d)} < 1$, this establishes the stated bound. $\qquad\square$

## B ROBUSTNESS CERTIFIABLE RADIUS

**Robustness Certifiable Radius.** Further, Lipschitz continuity is central to robustness certification. For a classifier $f$, the certified radius around input $x$ is lower-bounded as:

$$R(x) = \frac{F_y(x) - \max_{i \neq y} F_i(x)}{L_f(x)},$$

where $F_i(x)$ are the class logits, $y$ is the true label, $F_y(x)$ is the logit for the correct class, and $L_f(x)$ is the local Lipschitz constant. The numerator is the classification margin, and the denominator reflects input sensitivity.

The Lipschitz constant of the entire model can be upper-bounded by the product of the per-layer constants $L_f(x) \leq \prod_{i=1}^{L} L_f^{(i)}(x)$. Since we compare models that differ only in the attention layer, we treat all other $L_f^{(i)}(x)$ as fixed, isolating the contribution of the attention mechanism to the overall sensitivity. Thereby yielding the following theorem about the certified radius.

**Theorem 5.** Let $R_{\text{DA}}(x)$ and $R_{\text{base}}(x)$ be the certified radius for classifier $f$ incorporating DA and standard attention, respectively. Assume that all per-layer Lipschitz constants $L_f^{(i)}(x)$ except for the attention layer are identical between the two models, isolating the effect of the attention mechanism. Then, the ratio of certified radii satisfies:

$$\frac{R_{\text{DA}}(x)}{R_{\text{base}}(x)} \geq \frac{\Delta m}{\gamma\sqrt{1 + \lambda^2\rho^2 - 2\lambda\rho\cos\theta}} \tag{6}$$

where $m_{\text{DA}}$ is the classification margin under DA, and $\Delta m = \frac{m_{\text{DA}}}{m_{\text{base}}}$. In particular, under the opposite alignment $\cos\theta = -1$, the bound admits the asymptotic form $\mathcal{O}\Big(\frac{\Delta m}{\gamma(1+\lambda\rho)}\Big)$.

*Proof.* Now, using the definition of certified radius and margin ratio $\Delta m := \frac{m_{\text{DA}}}{m_{\text{base}}}$, we compute:

$$\frac{R_{\text{DA}}(x)}{R_{\text{base}}(x)} = \frac{m_{\text{DA}}/L_{\text{DA}}(x)}{m_{\text{base}}/L_{\text{base}}(x)} = \Delta m \cdot \frac{L_{\text{base}}(x)}{L_{\text{DA}}(x)}.$$

Then applying the Lipschitz ratio bound derived in Lemma 2, we finally have the claim (6). $\qquad\square$

This formulation illustrates a key trade-off in DA: while the subtractive mechanism can sharpen focus and modestly increase classification margins (i.e., $\Delta m > 1$), it also amplifies local sensitivity through the $\gamma\sqrt{1 + \lambda^2\rho^2 - 2\lambda\rho\cos\theta}$ factor—particularly under negative gradient alignment. Since

this amplification often outweighs the margin gain, the certified robustness deteriorates, as reflected in a shrinking certified radius.

Here, while $\gamma$ and $\lambda$ are fixed constants, both $\Delta m$, $\rho$ and $\theta$ depend on the perturbation direction $\xi$ and can thus be influenced by an adversary. This implies that an attacker may reduce the certified radius by manipulating $\rho$ (e.g., increasing gradient asymmetry) or suppressing the margin $m_{\text{DA}}$ through targeted perturbations.

## C    IMPLEMENTATION DETAILS

This section describes the implementation details of our experiments to ensure reproducibility.

### C.1    TARGET MODELS

DiffCLIP (Hammoud & Ghanem, 2025) extends CLIP (Radford et al., 2021) by replacing standard self-attention with Differential Attention (DA). It retains CLIP's original contrastive training objective between images and text, while using DA to suppress redundant activations and enhance focus on discriminative regions. Additionally, (Hammoud & Ghanem, 2025) introduces DiffViT, a vision transformer variant that incorporates DA into ViT architectures.

**DiffCLIP and CLIP.** We use the pre-trained weights of DiffCLIP_ViTB16_CC12M released by (Hammoud & Ghanem, 2025), which are trained on CC12M (Changpinyo et al., 2021) datasets. The authors have also publicly released the corresponding CLIP models via their HuggingFace repository.[1]

**DiffViT and ViT.** For ViT, we adopt the architecture provided in the `timm` library (Wightman, 2019) and train it on CIFAR-10 and CIFAR-100. We implement DiffViT following the design in (Hammoud & Ghanem, 2025) and train it on the same datasets. We train the models for 100 epochs on CIFAR-10 and 300 epochs on CIFAR-100 using the Adam optimizer with a learning rate of $\eta = 5 \times 10^{-4}$ and a batch size of $B = 128$.

### C.2    ADVERSARIAL ATTACKS

#### C.2.1    PGD

We evaluate adversarial robustness using both adversarial patch attacks and adversarial examples, optimized via Projected Gradient Descent (PGD) (Madry et al., 2018) under $\ell_\infty$ norm constraints.

For CLIP and DiffCLIP, the attack objective is to break the semantic alignment between visual and textual representations. We minimize the cosine similarity between the perturbed visual embedding and its associated text prompt:

$$\min_\delta \cos\left(\phi_{\text{visual}}(x + \delta), \phi_{\text{text}}(t)\right),$$

where $x$ is the input image, $t$ is the class prompt (e.g., "a photo of $<$class$>$"), and $\phi_{\text{visual}}, \phi_{\text{text}}$ denote the visual and text encoders, respectively.

For ViT and DiffViT on CIFAR-10/100, the attack instead minimizes the (negative) cross-entropy loss to induce misclassification.

In the adversarial patch setting, the perturbation $\delta$ is constrained to a fixed-size square patch of dimension $w \times w$ inserted at a random location in $x$. We set $\epsilon = 1.0$ and vary the patch size $w$.

#### C.2.2    AUTOATTACK

AutoAttack is an ensemble adversarial evaluation framework that combines four complementary attacks: APGD-CE, APGD-DLR, FAB (Croce & Hein, 2020a), and Square Attack (Andriushchenko et al., 2020). This combination provides a strong and diverse set of adversarial perturbations for robustness evaluation. We employ the official AutoAttack implementation available at GitHub[2].

---

[1]https://huggingface.co/collections/hammh0a/diffclip-67cd8d3b7c6e6ea1cc26cd93
[2]https://github.com/fra31/auto-attack

### C.2.3 CARLINI–WAGNER (CW) ATTACK

The Carlini–Wagner attack (Carlini & Wagner, 2017) is a strong optimization-based adversarial method that searches for minimal perturbations capable of inducing misclassification. Unlike iterative gradient-based methods such as PGD, CW directly formulates the attack as a constrained optimization problem under an $\ell_2$ norm. We use a PyTorch implementation based on the open-source repository[3].

## D ADDITIONAL EXPERIMENTAL RESULTS

### D.1 ASR ON TINYIMAGENET

On Tiny ImageNet (Table 2), we observe a consistent pattern: DiffViT shows higher ASR than ViT across patch sizes and small $\ell_\infty$ budgets. The gap is most pronounced at $\epsilon$=1/255. At larger budgets ($\epsilon$=4/255), the two models converge, echoing the cancellation–fragility tradeoff seen in CIFAR experiments.

Table 2: ASR on TinyImageNet under PGD Attacks

| Model | patch=8 | patch=4 | patch=2 | $\epsilon = 4/255$ | $\epsilon = 2/255$ | $\epsilon = 1/255$ | $\epsilon = 0.5/255$ |
|---|---|---|---|---|---|---|---|
| DiffViT | **0.6465** | **0.4556** | **0.1834** | 0.6318 | **0.3866** | **0.2822** | **0.2364** |
| ViT | 0.6139 | 0.4072 | 0.1498 | **0.6490** | 0.3857 | 0.2309 | 0.1866 |

### D.2 ADVERSARIAL TRAINING

In addition to balancing the structural trade-off inherent to DA, complementary robustness strategies can be applied. Adversarial training offers a complementary strategy that orthogonal to DA's subtractive structure.

Table 3 report the result. The adversarial training with small perturbations successfully reduced ASR without harming classification accuracy. However, beyond $\epsilon = 0.5/255$, it drastically drops the clean accuracy. Therefore, adversarial training is a candidate technique to increase the adversary robustness, but is not a silver-bullet.

Table 3: Attack success rates and clean accuracy of vanilla DiffViT and DiffViT with adversarial training when varying $\epsilon$ on CIFAR-10.

| | Accuracy | ASR ($\epsilon$=0.25/255) | ASR ($\epsilon$=0.5/255) | ASR ($\epsilon$=1/255) |
|---|---|---|---|---|
| DiffViT (CIFAR-10) | 0.8700 | 0.3310 | 0.6367 | 0.8498 |
| + Adv. Train ($\epsilon$=0.25/255) | 0.9518 | 0.2537 | 0.6434 | 0.8824 |
| + Adv. Train ($\epsilon$=0.5/255) | 0.8931 | 0.1787 | 0.4440 | 0.7676 |
| + Adv. Train ($\epsilon$=1/255) | 0.7384 | 0.0982 | 0.2089 | 0.4302 |

### D.3 PGD WITH RANDOM RESTART

To verify the stability of our PGD evaluations, we additionally performed PGD attacks with random restarts on all ViT/DiffViT and CLIP/DiffCLIP models. The original results in the main paper reported the average ASR over multiple runs, whereas the new experiments explicitly re-run PGD with three random initializations per input.

Figure 6 shows the result. Across all datasets (CIFAR-10/100, ImageNet, and COCO), random restart increases absolute ASR values—as expected from a stronger optimization procedure—but the *relative behavior* between DA and standard attention remains unchanged. In all settings, DA exhibits higher sensitivity for small perturbations, and the depth-dependent trends observed in the main paper are fully preserved.

---

[3]https://github.com/kkew3/pytorch-cw2

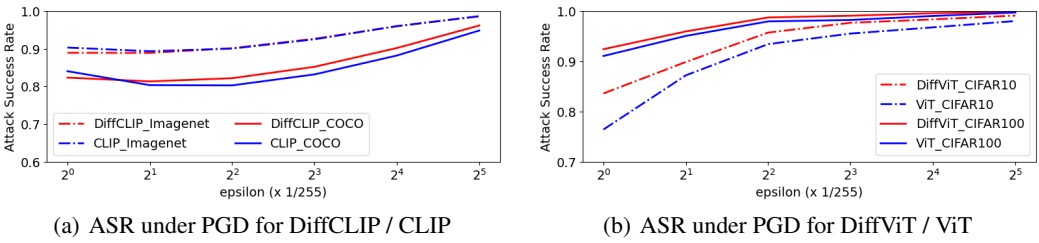

(a) ASR under PGD for DiffCLIP / CLIP    (b) ASR under PGD for DiffViT / ViT

Figure 6: ASR under PGD with Random Restart.

For ViT/DiffViT, random restart yields consistent increases in ASR as $\epsilon$ grows, with DiffViT remaining uniformly more vulnerable than ViT. For CLIP/DiffCLIP, evaluations on both ImageNet and COCO also show the same pattern: DiffCLIP maintains equal or higher ASR compared to CLIP across all perturbation budgets.

Overall, the random-restart results confirm that our conclusions are not an artifact of single-run PGD but remain robust under stronger multi-start attacks.

# E    LLM USAGE

We used large language models (LLMs) as writing assistants to improve clarity and readability. They were not involved in generating ideas, conducting experiments, or producing results. All technical content and contributions are solely by the authors.

