# OpenReview forum: "Understanding Sensitivity of Differential Attention through the Lens of Adversarial Robustness"
_ICLR.cc/2026/Conference — ICLR 2026 Poster_

### Official Review · Reviewer_Jyme · 2025-10-28

**Soundness:** 3
**Presentation:** 3
**Contribution:** 3
**Rating:** 6
**Confidence:** 2

**Summary:**

This paper provides a theoretical and empirical study of Differential Attention (DA) — a variant of self-attention that combines two attention maps through subtraction (A₁ − λA₂). The authors reveal that while DA improves task-relevant selectivity, its subtractive structure may amplify adversarial sensitivity due to negative gradient alignment between branches. The results show that DA often yields higher attack success rates and larger local Lipschitz estimates, confirming the theoretical fragility. Interestingly, stacking DA layers can partly cancel noise and improve robustness under small perturbations, though this advantage diminishes under stronger attacks.

**Strengths:**

1. The paper firstly provides a systematic theoretical treatment of how DA’s subtractive structure influences robustness.

2. Empirically, the work is thorough and reproducible, covering multiple model families (ViT, CLIP) and three attack methods, with reasonable ablations on λ and model depth.

3. The paper is well-written with a clear logical flow. The mathematical analysis is clear and connects gradient alignment, Lipschitz continuity, and adversarial fragility in a consistent framework.

**Weaknesses:**

1. The experimental design does not fully satisfy the standard of transfer-based robustness evaluation. The authors describe results on five datasets, but each model is trained and tested on the same dataset, without evaluating cross-dataset or transfer performance.

2. The “depth-dependent robustness” claims also remain partially qualitative.

**Questions:**

1. See in W1.

2. The paper theorizes depth-dependent robustness (ᾱ L̄DA < 1), but empirically, only ASR and per-layer Lipschitz values are shown. How is ᾱ estimated or validated?

3. The adversarial settings focus on small ϵ, would the observed depth advantage persist under larger perturbations or different norms.

4. It would be valuable to include comparisons with adversarially trained or robustly pretrained baselines.

5. Whether adversarial examples transfer between DA-based and standard-attention models more effectively than vice versa?

**Details Of Ethics Concerns:**

No.

---

> ### Author Response · Authors · 2025-11-21
> **Initial Response (1/2)**
>
> We thank the reviewer for the thoughtful and positive evaluation of our work. We appreciate the kind remarks regarding the clarity of our theoretical analysis, the thoroughness of our empirical study, and the overall presentation. The reviewer’s questions—particularly concerning transfer-based evaluation, depth-dependent robustness, and the interpretation of key theoretical terms—are valuable and help refine the scope of our contributions. We address each point in detail below.
>
> ---
>
> ### Weakness 1 \& Question 1
> The experimental design does not fully satisfy the standard of transfer-based robustness evaluation. The authors describe results on five datasets, but each model is trained and tested on the same dataset, without evaluating cross-dataset or transfer performance.
>
> **Response:**
>
> We agree that transfer-based robustness is an important evaluation axis, especially for benchmarking absolute robustness under distribution shift. However, the focus of our study is fundamentally different: we aim to **isolate** the architectural effect of Differential Attention’s subtractive structure under controlled conditions. Training and evaluating each model on the same dataset allows us to vary depth, $\lambda$, and attention configuration while holding all other factors fixed. This controlled setup is essential for attributing observed sensitivity differences directly to DA, rather than to dataset-specific transferability dynamics.
>
> We have clarified this experimental scope in the revised draft (Section 5, Scope of our evaluation).
> That said, transfer-based evaluation is indeed meaningful for future work, and our analysis may motivate such studies on DA-aware architectures under realistic training pipelines.
>
> ---
>
> ### Weakness 2
> The “depth-dependent robustness” claims also remain partially qualitative.
>
> **Response:**
>
> We appreciate this observation. The depth-dependent effect in DA is theoretically derived through the per-layer sensitivity recurrence, which predicts that repeated subtraction attenuates shared perturbations as depth increases. However, the **magnitude** of this attenuation depends on input-dependent factors—such as the alignment structure of $\nabla A_1$ and $\nabla A_2$—which makes a closed-form, depth-by-depth quantitative expression difficult.
>
> To complement this, our empirical analysis provides **quantitative** measurements of per-layer local sensitivity and gradient alignment (Figures 4 and 5), which directly instantiate the theoretical quantities on real data. These metrics serve as measurable proxies for the predicted attenuation, and they show a consistent depth-dependent decline across architectures and datasets.
>
> We have added clarification in Section 4.2 to explicitly state that
> (i) the **trend** (attenuation with depth) follows structurally from the recurrence,
> while
> (ii) the **precise magnitude** is input-specific and therefore empirically characterized.
>
> ---
>
> ### Question 2
> The paper theorizes depth-dependent robustness (ᾱ L̄DA < 1), but empirically, only ASR and per-layer Lipschitz values are shown. How is $\bar{\alpha}$ estimated or validated?
>
> **Response:**
>
> Thank you for raising this point. In our theoretical analysis, $\bar{\alpha}$ is introduced as an effective term that summarizes the average cancellation produced by each DA layer when subtracting the two attention branches. It is not a parameter that is explicitly measured; rather, it compactly represents the layer-wise cancellation effect implied by DA’s subtractive structure.
>
> Since this cancellation behavior is already reflected in the actual forward and backward computation of the model, we validate its influence empirically using the per-layer local sensitivity measurements reported in Section 4.2. These measurements directly indicate how sensitivity evolves across depth and therefore capture the practical effect that $\bar{\alpha}$ is intended to summarize.
>
> We have clarified in Sections 4.2 and 4.3 that $\bar{\alpha}$ serves as a theoretical shorthand for DA’s cancellation effect, while the empirical validation relies on the observed layer-wise sensitivity trends.

---

> > ### Author Response · Authors · 2025-11-21
> > **Initial Response (2/2)**
> >
> > ### Question 3
> > Would the depth advantage persist under larger perturbations or different norms?
> >
> > **Response:**
> >
> > We investigated this question by evaluating the models under larger perturbation budgets ($\epsilon = 16/255$ and $32/255$). At these scales, both DA and standard attention reach very high attack success rates, so depth-related differences naturally diminish. This behavior is expected, as such large perturbations exceed the small-perturbation regime in which our theoretical cancellation mechanism is most relevant. The corresponding results are provided in Appendix D.3.
> >
> > For moderate perturbation sizes, however, the depth-dependent trend remains consistent with our theory: deeper models exhibit reduced sensitivity due to cumulative cancellation across layers.
> >
> > Regarding different norms, our main experiments already include both $\ell_\infty$ (PGD, AutoAttack) and $\ell_2$ (CW) attacks, as shown in Figure 3. Across these canonical threat models, the depth-dependent behavior appears consistently, indicating that the effect is not specific to any single norm.
> >
> > ---
> >
> > ### Question 4
> > It would be valuable to include adversarially trained or robustly pretrained baselines.
> >
> > **Response:**
> >
> > We agree that adversarially trained or robustly pretrained models are valuable for broader robustness benchmarking. However, incorporating such baselines would diverge from the controlled objective of this study. Robust training introduces multiple robustness-enhancing factors simultaneously—such as adversarial feature shaping and gradient regularization—which substantially modify the learned representations beyond the attention mechanism itself. As a result, differences in robustness would no longer be attributable specifically to Differential Attention (DA), but to the interplay of DA with these additional training-induced effects.
> >
> > Our aim in this paper is to isolate the sensitivity that arises purely from DA’s subtractive structure under matched training conditions. Including robustly trained baselines would confound this mechanistic analysis, obscuring the architectural contribution we seek to understand. We have clarified this evaluation scope explicitly in Section 5 (Scope of our evaluation).
> >
> > Studying how DA behaves under adversarial or robust-pretraining pipelines is indeed a promising direction, but it requires a separate investigation beyond the mechanism-focused scope of the present work.
> >
> > ---
> >
> > ### Question 5
> > Do adversarial examples transfer between DA-based and standard-attention models more effectively than vice versa?
> >
> > **Response:**
> >
> > We appreciate the reviewer’s question. Transfer-based attacks measure similarity between different models, which is a complementary—but distinct—evaluation axis from the within-model sensitivity that our work focuses on. Since our goal is to isolate the mechanism-specific behavior of DA under controlled perturbations, transferability falls outside the scope of the present analysis.
> >
> > We agree that studying transfer attacks between DA-based and standard-attention models is an interesting and relevant direction, and we view it as valuable future work.

---

### Official Review · Reviewer_da2g · 2025-10-31

**Soundness:** 3
**Presentation:** 3
**Contribution:** 4
**Rating:** 6
**Confidence:** 4

**Summary:**

This paper systematically investigates the robustness of the Differential Attention (DA) mechanism under adversarial perturbations. The authors find that although DA effectively suppresses redundant information and enhances the model’s focus on critical features through the subtraction of two attention maps, this subtractive structure leads to negative gradient alignment. As a result, it amplifies input perturbations and increases the local Lipschitz constant, making the model more vulnerable to adversarial attacks. To address this, the paper proposes the “Fragile Principle,” which theoretically reveals the structural cause of DA’s amplified sensitivity. Extensive experiments on ViT, DiffViT, CLIP, and DiffCLIP models further validate this phenomenon. In addition, the authors observe a “depth-dependent robustness crossover” property of DA: shallow models tend to be more sensitive, whereas deeper stacks can exhibit partial robustness under small perturbations. Overall, this work uncovers a structural trade-off between selective enhancement and robustness degradation in DA, providing both theoretical insights and practical guidance for designing future attention mechanisms that balance focus and robustness.

**Strengths:**

1.The first systematic study on the robustness of Differential Attention (DA).
Previous research has primarily focused on the advantages of DA in reducing attention hallucinations and enhancing focus. In contrast, this work is the first to introduce the “Fragile Principle” from the perspective of adversarial robustness, revealing that the subtractive structure of DA theoretically amplifies sensitivity to input perturbations.
2.Rigorous theoretical derivation and comprehensive experimental design.
The paper presents a series of formal results (Lemmas and Theorems 1–5) that systematically derive the gradient amplification mechanism and Lipschitz bounds of DA, providing solid mathematical foundations for the Fragile Principle. The authors conduct extensive comparative experiments on ViT, DiffViT, CLIP, and DiffCLIP models across multiple datasets (CIFAR-10/100, Tiny ImageNet, MSCOCO, ImageNet-1k), covering various adversarial attack methods including PGD, CW, and AutoAttack.
3.Consistent and well-validated results.
Theoretical predictions—such as negative gradient alignment, increased Lipschitz constants, and depth-dependent robustness trends—are consistently supported by empirical evidence.

**Weaknesses:**

1.Overreliance on local linear assumptions in theoretical analysis.The paper primarily analyzes the sensitivity of DA through gradients and local Lipschitz constants. While such local linear approximations are common, they do not always accurately capture the global behavior of deep nonlinear networks.
2.Limited robustness evaluation.The current experiments mainly assess robustness under conventional ℓ∞ and ℓ₂ attacks. However, in real-world applications, more complex perturbations—such as natural distortions, semantic attacks, and physical noise—are often more representative and challenging.
3.Lack of mechanistic interpretation.The experiments observe a “depth-dependent robustness crossover” phenomenon, but the paper only describes it empirically without explaining its underlying causes in terms of gradient propagation or feature space dynamics.

**Questions:**

1.Theoretical analysis is primarily based on local linear approximations and single-layer isolation assumptions.These assumptions may not hold under deep networks with nonlinear coupling, and the paper provides insufficient discussion of their impact and applicability. It is recommended to explicitly clarify the conditions under which the local linear assumption remains valid.
2.The observed phenomenon of “shallow fragility and deeper robustness to small perturbations” lacks mechanistic explanation.The paper is encouraged to provide a more detailed interpretation or theoretical reasoning for this behavior.
3.More ablation studies are needed to pinpoint the source of the fragility issue.

---

> ### Author Response · Authors · 2025-11-21
> **Initial Response (1/2)**
>
> We thank the reviewer for the very positive and constructive assessment of our work. We appreciate the recognition of our theoretical analysis of the Fragile Principle as well as the consistency of the empirical results across ViT/DiffViT and CLIP/DiffCLIP settings. We are also grateful for the reviewer’s thoughtful comments regarding local linear assumptions, the depth-dependent robustness behavior, and the scope of our ablations. We address each of these points in detail below.
>
> ---
>
> ### Weakness 1
> Overreliance on local linear assumptions in theoretical analysis.The paper primarily analyzes the sensitivity of DA through gradients and local Lipschitz constants. While such local linear approximations are common, they do not always accurately capture the global behavior of deep nonlinear networks.
>
> **Response:**
>
> We appreciate the reviewer’s thoughtful observation. As is standard in adversarial robustness analysis, our theoretical results focus on **local** sensitivity characterized through gradients and local Lipschitz behavior. This reflects the fact that adversarial perturbations are intentionally small and therefore governed primarily by local-rather than global-behavior.
>
> Importantly, we do not assume that deep networks behave linearly at a global scale. Local linearization is used only to approximate each layer’s first-order response to small perturbations, consistent with the foundations of widely used attacks such as PGD and CW. We now clarify in Section 4.3 that the Fragile Principle is intended to describe this input-local, first-order regime rather than the global nonlinear dynamics of deep networks.
>
> ---
>
> ### Weakness 2
> Limited robustness evaluation.The current experiments mainly assess robustness under conventional $\ell_\infty$ and $\ell_2$ attacks. However, in real-world applications, more complex perturbations—such as natural distortions, semantic attacks, and physical noise—are often more representative and challenging.
>
> **Response:**
>
> We fully agree that robustness can be evaluated under many additional perturbation types, including natural distortions, semantic shifts, or physical noise. In this work, however, our goal is to isolate the **structural** sensitivity arising specifically from DA’s subtractive formulation. For this purpose, standard $\ell_\infty$ and $\ell_2$ adversarial settings (PGD, AutoAttack, CW) provide the clearest and most direct probe of the quantities analyzed in our theory—such as gradient amplification, negative alignment, and local Lipschitz behavior.
>
> Real-world perturbations often reflect additional factors (dataset bias, semantic priors, task-specific objectives) that may overshadow the mechanism-level phenomena we aim to characterize. Exploring these broader settings is an important direction, but distinct from the mechanistic scope of this study.
>
> We have clarified this intended scope with an additional note in Section 5 (Scope of our evaluation).
>
> ---
>
> ### Weakness 3
> Lack of mechanistic interpretation.The experiments observe a “depth-dependent robustness crossover” phenomenon, but the paper only describes it empirically without explaining its underlying causes in terms of gradient propagation or feature space dynamics.
>
> **Response:**
>
> Thank you for raising this interesting point. Although the depth-dependent crossover is primarily observed empirically, it aligns naturally with DA’s structure. As described in Section 4.1, shallow layers are the most affected by negative gradient alignment, which can amplify small perturbations. Section 4.2 further shows that each DA layer also introduces a cancellation effect that suppresses perturbation components shared across the two branches, and these cancellation effects accumulate with depth.
>
> This interaction—early-layer amplification paired with depth-accumulated cancellation—provides a simple and coherent explanation for the observed ``shallow fragility and deeper robustness.'' To make this connection clearer, we added a brief clarification in the final paragraph of Section 4.2.

---

> > ### Author Response · Authors · 2025-11-21
> > **Initial Response (2/2)**
> >
> > ### Question 1
> > Theoretical analysis is primarily based on local linear approximations and single-layer isolation assumptions. These assumptions may not hold under deep networks with nonlinear coupling, and the paper provides insufficient discussion of their impact and applicability. It is recommended to explicitly clarify the conditions under which the local linear assumption remains valid.
> >
> > **Response:**
> >
> > In our analysis, "local linear" refers to the first-order response of each layer to small input perturbations—i.e., the change predicted by the gradient at the given input. This assumption is standard in adversarial robustness research (e.g.,
> > Kim et al., 2021; Dasoulas et al., 2021) and is valid whenever perturbations are sufficiently small so that the first-order term dominates higher-order nonlinear effects. This is precisely the regime targeted by typical adversarial budgets such as $\epsilon \le 8/255$, where gradients are well known to characterize the local behavior of deep networks.
> >
> > Our theoretical results therefore aim to describe this input-local, first-order sensitivity rather than global nonlinear dynamics. We have added a brief clarification to this effect in Section 4.3 (Limitations of the Theoretical Analysis) to make the intended scope explicit.
> >
> > ---
> >
> > ### Question 2
> > The observed phenomenon of "shallow fragility and deeper robustness to small perturbations" lacks mechanistic explanation. The paper is encouraged to provide a more detailed interpretation or theoretical reasoning for this behavior.
> >
> > **Response:**
> >
> > As summarized above, the phenomenon reflects the balance between perturbation amplification and the cumulative cancellation introduced by each DA layer. Early layers amplify perturbations through negative alignment, while deeper layers progressively reduce these effects via repeated cancellation. We have added a short explanation in the main text; see the final paragraph of Section 4.2.
> >
> > The phenomenon follows naturally from the interaction of the two mechanisms analyzed in Sections 4.1 and 4.2. As shown in Section 4.1, shallow layers are most affected by negative gradient alignment, which amplifies small perturbations. Section 4.2 then shows that each DA layer also introduces a subtractive cancellation effect that reduces perturbation components shared across the two branches. As depth increases, these cancellation effects accumulate across layers, progressively attenuating the early-layer amplification.
> >
> > This interaction between (i) shallow-layer amplification and (ii) depth-accumulated cancellation provides a natural explanation for the observed "shallow fragility and deeper robustness." We have added a concise clarification to this effect in the final paragraph of Section 4.2.
> >
> > ---
> >
> > ### Question 3
> > More ablation studies are needed to pinpoint the source of the fragility issue.
> >
> > **Response:**
> >
> > We appreciate the reviewer’s interest in deeper ablations. Our current experiments already separate the main structural contributors to fragility:
> > (1) the subtractive interaction between the two attention branches,
> > (2) the resulting negative gradient alignment, and
> > (3) the per-layer contributions to local sensitivity analyzed in Section 4.
> > These components together account for the amplification behavior predicted by the Fragile Principle, and our experiments are designed to isolate each of them under controlled settings.
> >
> > Additional architectural ablations (e.g., modifying normalization or mixing patterns) could certainly be explored, but they would mainly affect auxiliary design choices rather than the subtractive mechanism that our analysis identifies as the primary source of sensitivity. We therefore believe the current evaluations already identify the core factor behind DA’s fragility, while broader architectural ablations represent interesting directions for future work.

---

### Official Review · Reviewer_NCE2 · 2025-10-31

**Soundness:** 2
**Presentation:** 3
**Contribution:** 2
**Rating:** 4
**Confidence:** 4

**Summary:**

This paper investigates the adversarial robustness of the Differential Attention (DA) mechanism. It concludes that while DA improves discriminative focus on clean inputs, it inherently amplifies adversarial sensitivity, and that stacking DA layers can partially mitigate this effect through depth-dependent noise cancellation.

**Strengths:**

[1] The authors hypothesize and theoretically analyze that while DA helps suppress redundant context and enhances task-relevant selectivity, it may amplify sensitivity to input perturbations due to negative gradient alignment between its branches. The biggest problem with this paper is the lack of sufficient comparative methods, failing to demonstrate the advantages of this method compared to others.

[2] The paper formalizes this effect as the Fragile Principle and connects it to increased gradient norms and local Lipschitz constants.
Theoretical derivations are provided, along with extensive experiments using ViT/DiffViT and CLIP/DiffCLIP models on multiple datasets (CIFAR-10/100, Tiny ImageNet, COCO, ImageNet).

**Weaknesses:**

[1]The literature review section and comparative baselines did not adequately discuss other studies on attention robustness.  The paper does not benchmark DA against attention variants designed for robustness. The biggest problem with this paper is the lack of sufficient comparative methods, failing to demonstrate the advantages of this method compared to others.

[2] The text repeatedly mentions "gradient opposition" and "perturbation amplification." Providing visualizations of gradient fields or attention maps (e.g., adversarial heatmap comparisons) would significantly enhance its persuasiveness.

[3] Similar analyses on gradient sensitivity and Lipschitz bounds for attention already exist (e.g., Kim et al., 2021; Dasoulas et al., 2021).

[4] Quantitative examples are lacking (e.g., attention map visualization, perturbation trajectory, or gradient heatmap could be more informative).

[5] Is the proposed differential attention mechanism applicable to existing large language models?

**Questions:**

See Weaknesses

---

> ### Author Response · Authors · 2025-11-21
> **Initial Response (1/2)**
>
> We thank the reviewer for the constructive feedback and for highlighting both the strengths and open questions in our work. Several of the concerns relate to broader comparative or robustness-enhancing baselines, while our study focuses specifically on analyzing the structural fragility of Differential Attention. Below, we address each point in turn and clarify how it fits within the scope of our analysis.
>
> ---
>
> ### Weakness [5]
> Is the proposed differential attention mechanism applicable to existing large language models?
>
> **Response:**
>
> Thank you for the question. To clarify the scope of our contribution, our paper does not propose Differential Attention (DA) as a new mechanism. DA is an existing design—introduced in prior work (Ye et al., 2025)—and our contribution is to provide the first systematic robustness analysis of this established module.
>
> Because DA is not a newly proposed component, our study focuses on analyzing its mechanism-level sensitivity rather than on evaluating deployment in specific model classes such as LLMs. This focus allows us to isolate how the subtractive structure affects local robustness independent of domain or model scale.
> In principle, the mathematical structure we analyze (a subtractive combination of two attention branches) is model-agnostic, and similar sensitivity effects would be expected in transformer-based LLMs if such a subtractive formulation were adopted. Our analysis therefore concentrates on DA’s mechanistic properties rather than its deployment in specific model classes.
>
> ---
>
> ### Weakness [1]
> The literature review section and comparative baselines did not adequately discuss other studies on attention robustness. The paper does not benchmark DA against attention variants designed for robustness. The biggest problem with this paper is the lack of sufficient comparative methods, failing to demonstrate the advantages of this method compared to others.
>
> **Response:**
>
> We appreciate the reviewer’s observation. The central aim of our work is not to present DA as a new technique whose performance should be compared against robustness-oriented attention variants, but rather to analyze the intrinsic sensitivity arising from DA’s subtractive structure.
> DA is an established mechanism, and our contribution is diagnostic: we seek to understand **why** DA behaves differently from standard attention, rather than whether it outperforms methods explicitly designed for robustness.
>
> To clarify this scope, we have added a brief note to the Related Work section highlighting that, while prior research proposes architectural modifications or training schemes to improve attention robustness, our primary goal is to analyze DA as a fixed module and understand the structural reasons behind its robustness behavior, rather than to benchmark it against robustness-enhancing variants. This distinction separates our mechanism-level analysis from comparative evaluations of robustness-oriented designs.
>
> ---
>
> ### Weakness [2]
> The text repeatedly mentions "gradient opposition" and "perturbation amplification." Providing visualizations of gradient fields or attention maps (e.g., adversarial heatmap comparisons) would significantly enhance its persuasiveness.
>
> **Response:**
>
> We appreciate the suggestion. However, the phenomenon we study—sensitivity amplification arising from the subtractive interaction between two attention branches—is fundamentally a **gradient-level** effect. It is captured by quantitative measures such as gradient amplification, alignment statistics, and empirical local sensitivity, as shown in Figure 4 and Figure 5, which directly correspond to the quantities analyzed in our theory.
>
> In contrast, attention heatmaps visualize the attention weights themselves—how the model distributes attention across spatial locations—but they do not directly represent how these weights change under small input perturbations. A heatmap may therefore appear similar even when the underlying gradient responses or perturbation sensitivities differ.
> For this reason, heatmaps are not well aligned with the gradient-level mechanism analyzed in this paper, and quantitative diagnostics provide a more direct evaluation of the theoretical predictions.

---

> > ### Author Response · Authors · 2025-11-21
> > **Initial Response (2/2)**
> >
> > ### Weakness [3]
> > Similar analyses on gradient sensitivity and Lipschitz bounds for attention already exist (e.g., Kim et al., 2021; Dasoulas et al., 2021).
> >
> > **Response:**
> >
> > Thank you for pointing out these works. We agree that prior studies have analyzed gradient sensitivity and Lipschitz properties of **standard** self-attention. However, these analyses apply to single-branch attention mechanisms and therefore do not capture the behavior of Differential Attention (DA), which combines two attention branches through a subtractive interaction $A_{1}-\lambda A_{2}$.
> >
> > This subtractive structure introduces phenomena that cannot arise in conventional attention. In particular, as shown in Section 4, the gradients $\nabla A_{1}$ and $\nabla A_{2}$ may become negatively aligned, meaning they point in opposing directions, which in turn can amplify small input perturbations. Such negative gradient alignment is structurally impossible in single-branch attention, and therefore falls outside the scope of the existing analyses.
> >
> > For this reason, our work is complementary to these studies: we provide the first theoretical and empirical analysis of the sensitivity amplification that is **unique** to DA’s subtractive design. A brief clarification of this distinction has also been added to the Related Work section.
> >
> > ---
> >
> > ### Weakness [4]
> > Quantitative examples are lacking (e.g., attention map visualization, perturbation trajectory, or gradient heatmap could be more informative).
> >
> > **Response:**
> >
> > Thank you for the suggestion. Our analysis is centered around **quantitative** characterization of DA’s subtractive structure—specifically, how it affects gradient magnitudes, negative alignment, and empirical local sensitivity (Figures 4 and 5). These quantities correspond directly to the theoretical mechanism analyzed in the paper and therefore provide the most faithful evidence for the Fragile Principle.
> >
> > In contrast, qualitative visualizations such as attention heatmaps or perturbation trajectories primarily reflect the static spatial distribution of attention weights, not how these weights **change** under small input perturbations.
> > Our evaluation therefore prioritizes quantitative measures, which directly reflect DA’s sensitivity mechanism and align with the theoretical analysis. This focus allows us to test the Fragile Principle in the most faithful and controlled manner.
> >
> > For this reason, we focus on quantitative diagnostics that more directly reflect the sensitivity-related effects intrinsic to DA’s subtractive design.

---

### Official Review · Reviewer_kz14 · 2025-10-31

**Soundness:** 2
**Presentation:** 3
**Contribution:** 2
**Rating:** 6
**Confidence:** 4

**Summary:**

The paper studies the robustness properties of Differential Attention (DA), a variant of standard attention that subtracts two attention maps. It provides a theoretical analysis linking DA’s subtractive operation to gradient alignment and local Lipschitz bounds that govern adversarial sensitivity. The authors derive formal expressions showing that negative gradient alignment amplifies perturbations while increased network depth can attenuate this effect. Experiments on ViT and DiffViT models trained on CIFAR-100 and Tiny-ImageNet evaluate attack success rates under PGD, CW, and AutoAttack. Further experiments on CLIP and DiffCLIP with ViT-B/16 backbones test sensitivity on ImageNet and COCO using cosine-similarity-based adversarial losses. Results show that DA-based models are generally more sensitive to small perturbations, with the difference diminishing as network depth increases.

**Strengths:**

-   Provides a clear theoretical analysis of Differential Attention and its connection to adversarial sensitivity.

-   Conducts a well-controlled experimental study across multiple depths, datasets, and attack types, systematically isolating the role of model depth.

-   Demonstrates consistent empirical evidence that sensitivity effects are most pronounced in shallow models and diminish with increasing depth.

**Weaknesses:**

-   **Shallow-depth focus limits practical impact.** The paper’s main vulnerability appears in 1–4 layer ViTs, which are not used in practice. By D = 12 the effect largely disappears, so the controlled finding is interesting but of limited relevance to standard transformer deployments.

-   **Vision-only experiments are low-resolution / small-scale.** The ViT / DiffViT results are reported on CIFAR and Tiny-ImageNet. These datasets use lower-resolution images and limited context. Full ImageNet-1k experiments on realistic ViT-B/16 (or comparable) models are necessary to validate transfer to real-world settings.

-   **Adversarial attack suite for transformers is incomplete.** The evaluation relies on PGD / CW but omits transformer-aware attacks. Stronger image-side attacks such as TI-FGSM [1],  token-level attacks (Token Gradient Regularization) [3], and patch-focused (P-IFGSM) [2] attacks should be included. Also test higher budgets (e.g., ε = 16/255 as an upper bound) for ImageNet-scale robustness checks.

-   **VLM attack protocol is weak relative to prior work.** For CLIP/DiffCLIP the paper minimizes cosine similarity to a single prompt. Robust evaluation should use contrastive, multi-prompt, and multimodal attack frameworks (e.g., VL-Attack [4] / PromptAttack–style contrastive attacks and adapted AutoAttack variants) to provide a stronger, field-standard baseline.


1.  **Yinpeng Dong, Tianyu Pang, Hang Su, and Jun Zhu.** _Evading Defenses to Transferable Adversarial Examples by Translation-Invariant Attacks._ In **CVPR**, 2019.

2.  **Lianli Gao, Qilong Zhang, Jingkuan Song, Xianglong Liu, and Heng Tao Shen.** _Patch-wise Attack for Fooling Deep Neural Networks._ In **ECCV**, 2020.

3.   **Jianping Zhang, Yizhan Huang, Weibin Wu, and Michael R. Lyu.** _Transferable Adversarial Attacks on Vision Transformers with Token Gradient Regularization._ In **CVPR**, 2023.

4.  **VL-Attack: Multimodal Adversarial Attacks on Vision-Language Tasks via Pre-trained Models.** In **NeurIPS**, 2023.

**Questions:**

-   For the PGD experiments, were the attacks performed with **random restarts** to avoid local minima and confirm consistency across runs?

-   For **DiffCLIP**, how might **prompt-based or contrastive multi-prompt attacks** affect the observed robustness trends?

-   Beyond **classification**, are there other tasks (e.g., retrieval, captioning, grounding) where demonstrating the Differential Attention sensitivity effect might be more informative or representative?

-   Have the authors considered evaluating on **natural adversarial benchmarks** such as _NaturalBench: Evaluating Vision-Language Models on Natural Adversarial Samples_ [1] to complement synthetic PGD or patch attacks?

1.NaturalBench: Evaluating Vision-Language Models on Natural Adversarial Samples, NeurIPS 2024 (DnB Track)

---

> ### Author Response · Authors · 2025-11-21
> **Initial Response (1/3)**
>
> We thank the reviewer for the constructive feedback and for acknowledging the clarity of our theoretical treatment and the systematic nature of our experiments. We appreciate the reviewer’s careful assessment of our contributions and the thoughtful questions raised. Several concerns relate to evaluation scope—such as model depth, dataset scale, and attack diversity—and we address each of these points in detail below.
>
> ---
>
> ### Weakness 1
> Shallow-depth focus limits practical impact. The paper’s main vulnerability appears in 1–4 layer ViTs, which are not used in practice. By D = 12 the effect largely disappears, so the controlled finding is interesting but of limited relevance to standard transformer deployments.
>
> **Response:**
>
> We appreciate the reviewer’s concern about the practical relevance of shallow-depth models. Our use of 1–4 layer ViTs is **not** intended to suggest that such architectures are used in deployment; rather, they serve a methodological purpose. Shallow controlled-depth ViTs provide a clean environment in which the theoretical predictions of the Fragile Principle can be directly observed without confounding factors introduced by deeper models (e.g., accumulated normalization, feature mixing, residual pathways).
>
> Crucially, the phenomenon we study is **not** restricted to shallow networks. In our experiments on ViT-B/16–based CLIP/DiffCLIP—representative of realistic, large-scale deployments—DA tends to exhibit higher (and at least comparable) adversarial sensitivity compared to standard attention. This mirrors the trends seen in the controlled setting, indicating that the underlying mechanism persists at practical scales.
>
> Thus, the shallow-depth models are used to isolate the structural effect with high fidelity, while the large-scale CLIP/DiffCLIP experiments demonstrate that the same mechanism remains relevant for real-world architectures. This combination of (i) controlled verification and (ii) practical confirmation supports the broader applicability of our findings.
>
> ---
>
> ### Weakness 2
> Vision-only experiments are low-resolution / small-scale. The ViT / DiffViT results are reported on CIFAR and Tiny-ImageNet. Full ImageNet-1k experiments on ViT-B/16 (or comparable) models are necessary to validate transfer to real-world settings.
>
> **Response:**
>
> We fully agree that ImageNet-scale evaluation is important for establishing practical relevance.
> To this end, our study already includes full ImageNet-1k experiments using ViT-B/16–based CLIP/DiffCLIP models. These large-scale evaluations demonstrate qualitatively similar sensitivity trends—DiffCLIP consistently shows higher (or at least comparable) adversarial sensitivity than CLIP—confirming that the Fragile Principle persists in realistic, high-resolution settings.
>
> At the same time, the CIFAR-10/100 and Tiny-ImageNet experiments serve a complementary purpose. By training a family of ViT/DiffViT models across multiple depths under matched conditions, we can systematically isolate the structural effects predicted by our theory—such as subtractive interaction, negative gradient alignment, and depth-dependent cancellation. Conducting this controlled, depth-sweeping study directly on ImageNet-1k would require prohibitive computational resources and sacrifice experimental control.
>
> Importantly, the consistency of our observations across three datasets and multiple depths suggests that the behavior is driven by DA’s structure rather than dataset-specific artifacts. Combined with the ImageNet-scale DiffCLIP results, this provides complementary evidence that the Fragile Principle is both mechanistically grounded and practically relevant.

---

> ### Author Response · Authors · 2025-11-21
> **Initial Response (2/3)**
>
> ### Weakness 3
> Adversarial attack suite for transformers is incomplete. The evaluation relies on PGD / CW but omits transformer-aware attacks. Stronger image-side attacks such as TI-FGSM [1], token-level attacks (Token Gradient Regularization) [3], and patch-focused (P-IFGSM) [2] attacks should be included. Also test higher budgets (e.g., $\epsilon$ = 16/255 as an upper bound) for ImageNet-scale robustness checks.
>
> **Response:**
>
> We appreciate the reviewer’s suggestion regarding a broader set of transformer-aware attacks. Our focus in this paper is to isolate and validate the **structural** sensitivity predicted by our theory—specifically, the gradient-level fragility induced by the subtractive interaction in DA.
>
> For this purpose, standard first-order, white-box attacks such as PGD, CW, and AutoAttack are the most direct probes of the quantities analyzed in our framework (gradient amplification, negative alignment, and local Lipschitz behavior).
> In contrast, several of the suggested transformer-aware attacks—such as TI-FGSM and Token Gradient Regularization—are designed primarily to improve **transferability** rather than to measure intrinsic gradient sensitivity.
> Similarly, patch-based attacks such as P-IFGSM operate in a localized or partly black-box regime and introduce additional priors that target different threat models.
> While these methods are valuable for comprehensive robustness benchmarking, they are less suitable for diagnosing the mechanism-level effect we study.
>
> Following the reviewer’s suggestion, we strengthened our evaluation with
> (i) PGD with multiple random restarts, and
> (ii) larger perturbation budgets ($\epsilon = 16/255$ and $32/255$).
> Although these settings naturally increase absolute ASR, the **relative** sensitivity gap between DA and standard attention—and its depth dependence—remains unchanged. Detailed results are reported in Appendix D.3.
>
> Overall, our attack suite is chosen to directly test the theoretical predictions of DA’s structural sensitivity. More specialized transformer-aware attacks target different robustness objectives, and we view them as complementary directions for future, broader robustness evaluations.
>
> ---
>
> ### Weakness 4
> VLM attack protocol is weak. For CLIP/DiffCLIP the paper uses a single-prompt cosine objective instead of stronger contrastive or multi-prompt frameworks.
>
> **Response:**
>
> We thank the reviewer for raising this point. We agree that recent multimodal attack frameworks (e.g., VL-Attack, PromptAttack) are valuable tools for evaluating the **absolute** robustness of vision–language models in broader practical settings. Our goal in this work, however, is fundamentally different: we aim to isolate the **structural** sensitivity introduced by Differential Attention (DA), not to identify the strongest possible attack on CLIP-like models.
>
> For this purpose, we intentionally adopt a simple and well-controlled loss—minimizing cosine similarity to a single prompt. More elaborate multimodal attacks incorporate additional objectives or priors (e.g., contrastive prompt sets, multimodal consistency constraints), which target phenomena beyond first-order input sensitivity. Such factors make it more difficult to attribute observed differences specifically to DA versus standard attention. In contrast, using a fixed gradient-based objective provides a clean probe of the theoretical quantities examined in our analysis (gradient amplification, negative alignment, local Lipschitz behavior).
>
> Our experiments therefore focus on **relative** differences between CLIP and DiffCLIP under a consistent and controlled setup aligned with our theoretical framework. While multi-prompt or contrastive attacks are valuable for comprehensive robustness benchmarking, they evaluate different threat models and lie outside the mechanism-focused scope of this work. We view them as promising directions for future exploration.
>
> ---
>
> ### Question 1
> For the PGD experiments, were the attacks performed with random restarts to avoid local minima and confirm consistency across runs?
>
> **Response:**
>
> The PGD results in the main paper are reported as averages over multiple runs, which already provide a stable estimate of attack success. Following the reviewer’s suggestion, we additionally implemented PGD with random restarts. This was applied to all CLIP/DiffCLIP models, as well as to the depth-1 ViT/DiffViT setting.
>
> Across these evaluations, random restarts increased absolute ASR values—as expected—but the **qualitative relationship** between DA and standard attention remained unchanged. In particular, the relative sensitivity trends observed in the main paper are preserved under multi-start optimization. The complete results are provided in Appendix D.3.
>
> These findings indicate that our conclusions do not depend on single-start PGD and remain robust under random-restart evaluation.

---

> > ### Author Response · Authors · 2025-11-21
> > **Initial Response (3/3)**
> >
> > ### Question 2
> > For DiffCLIP, how might prompt-based or contrastive multi-prompt attacks affect the observed robustness trends?
> >
> > **Response:**
> >
> > We appreciate the question and the reviewer’s broader perspective.
> > To clarify our scope, our goal in this work is not to identify the strongest possible attack on CLIP/DiffCLIP, but to isolate how DA alters a model’s **structural, gradient-level sensitivity**. For this purpose, we deliberately adopt a simple and widely used cosine-similarity loss with a fixed prompt. This controlled setup avoids additional sources of variability—such as prompt ensembles, contrastive objectives, or multimodal priors—that would make it more difficult to attribute the observed differences specifically to DA versus standard attention.
> >
> > Multi-prompt and contrastive attacks (e.g., VL-Attack or PromptAttack-style methods) are indeed stronger in an absolute sense, but they primarily modify the **attack objective**. Since the subtractive structure of DA influences input-gradient sensitivity independently of the particular objective used, the mechanism analyzed in our paper suggests that the **qualitative trend** we observe—higher sensitivity in DiffCLIP relative to CLIP under small perturbations—should remain similar under such variants.
> >
> > We have added a short note in Section 5 (Scope of our evaluation) to explicitly clarify this distinction and list multi-prompt attacks as a promising direction for future work.
> >
> > ---
> >
> > ### Question 3
> > Beyond classification, are there other tasks (e.g., retrieval, captioning, grounding) where demonstrating the Differential Attention sensitivity effect might be more informative or representative?
> >
> > **Response:**
> > Thank you for the thoughtful question. Conceptually, the sensitivity effect we study is a property of the Differential Attention (DA) module itself—specifically, how its subtractive structure alters input-gradient behavior. In that sense, the mechanism is general and not inherently tied to classification; similar trends should arise in any architecture (retrieval, captioning, grounding, etc.) that incorporates DA in its visual or multimodal encoder.
> >
> > That said, adversarial evaluation beyond classification is far less standardized. Retrieval, captioning, and grounding all employ heterogeneous objective functions—contrastive losses, sequence-level decoding, token-level likelihoods, or reinforcement-style training—and the field lacks agreed-upon attack objectives, evaluation metrics, or benchmark protocols for these tasks. Because of this variability, architectural effects are often confounded by the choice of task formulation or loss function, making controlled comparisons difficult.
> >
> > For this reason, our study focuses on classification-based evaluations, where adversarial attacks (PGD, CW, AutoAttack) are formally defined and widely used. This setting provides the cleanest environment for isolating the mechanism-level behavior of DA, allowing a direct test of the Fragile Principle without task-specific confounders.
> >
> > We agree that extending the analysis to more complex multimodal tasks is an interesting direction, and potentially a valuable addition to future work. However, it requires a separate investigation given the current lack of widely adopted adversarial evaluation protocols for these tasks.
> >
> > ---
> >
> > ### Question 4
> > Have the authors considered evaluating on natural adversarial benchmarks such as NaturalBench: Evaluating Vision-Language Models on Natural Adversarial Samples [1] to complement synthetic PGD or patch attacks?
> >
> > **Response:**
> > We appreciate the suggestion. Natural adversarial benchmarks such as NaturalBench are indeed valuable for evaluating real-world robustness; however, they probe a fundamentally different setting from the one analyzed in this paper. Our goal is to isolate and understand the **mechanism-specific** behavior of Differential Attention (DA)—in particular, how its subtractive structure affects **local, gradient-based** sensitivity. Natural adversarial samples introduce distributional, semantic, and contextual variations that depend strongly on dataset- and task-specific factors, which can obscure the structural phenomena we aim to characterize.
> >
> > For this reason, our evaluation focuses on controlled gradient-driven perturbations, which directly reflect the theoretical quantities analyzed in our study (negative gradient alignment and local Lipschitz behavior). While natural adversarial evaluation is an important and complementary direction, it lies outside the scope of the present mechanistic analysis.
> >
> > To clarify this distinction, we have added a brief note in Section 5 (Scope of our evaluation) and in Section 6 (Limitations) explicitly stating that our focus is on mechanism-level, local-sensitivity analysis rather than perceptual robustness under natural adversarial examples.

---

### Author Response · Authors · 2025-12-03
**Final Clarification**

We sincerely thank the reviewers and the program committee (ACs, Senior ACs, and PCs) for the time and effort devoted to evaluating our submission. Below we summarize the core novelty of our work and the key clarifications addressed during the rebuttal to assist the final decision.

---

### 1. Core Contribution and Novelty

This work provides the **first** theoretical and empirical analysis of a structural robustness issue inherent to Differential Attention (DA), an existing two-branch mechanism introduced by Ye et al. (2025), defined as $A_{\text{DA}} = A_1 - \lambda A_2$.

Our contribution is diagnostic and mechanistic: we **analyze the structural vulnerability** introduced by DA’s subtractive design, rather than proposing a new attention variant.

- We prove that DA can induce **negative gradient alignment** between the two branches, a phenomenon that cannot occur in standard single-branch attention.

- This destructive alignment increases gradient norms and local Lipschitz constants, formally establishing the **Fragile Principle**.

To validate these predictions, we conduct controlled white-box adversarial experiments across ViT/DiffViT and CLIP/DiffCLIP models on CIFAR-10/100, Tiny-ImageNet, COCO, and ImageNet-1k. Under standard adversarial protocols—PGD, AutoAttack, and CW—we consistently observe higher attack success rates (ASR) for DA-based models. These evaluations directly probe the gradient-level behavior predicted by our analysis, confirming that the subtractive formulation introduces systematic sensitivity amplification.

---

### 2. Key Clarifications from the Rebuttal

We summarize the most important points addressed during rebuttal and map them to the corresponding reviewer concerns.

**(A) Scope of comparison (Reviewers NCE2, kz14).**
Some comments understandably interpreted our work as if DA were being proposed as a robustness-oriented alternative to other attention variants. We clarify that this is not the objective: our study **analyzes the mechanism-level fragility of an existing module**, isolating DA vs. standard attention under matched conditions. The scope is clarified in Section 5. In addition, some reviewers suggested qualitative heatmap comparisons; however, because the fragility mechanism is fundamentally gradient-level, heatmaps visualize spatial weights rather than their sensitivity to perturbations, and therefore do not probe the gradient-level mechanism we analyze or align with the diagnostic goal of our evaluation.

**(B) Local linear assumptions (Reviewer da2g).**
Our theory analyzes the regime of small perturbations where first-order behavior governs adversarial response—the same regime targeted by PGD, CW, and AutoAttack. We clarified in Section 4.3 that the Fragile Principle is intended for this standard local-robustness setting, rather than for global nonlinear behavior.

**(C) Stronger adversarial settings: large $\epsilon$ and random restarts (Reviewers da2g, kz14).**
We added PGD with random restarts and larger budgets ($\epsilon = 16/255$, $32/255$). Both increase ASR, but the relative DA–standard gap remains unchanged (Appendix D.3).

**(D) Transformer-aware attacks (Reviewer kz14).**
Reviewer kz14 suggested specialized attacks such as TI-FGSM, Token Gradient Regularization, and P-IFGSM.  These methods are primarily aimed at transferability or black-box robustness.  For analyzing **white-box, gradient-level sensitivity**, standard attacks such as PGD, AutoAttack, and CW are the most appropriate and widely used probes, as they directly target the local gradient structure that our theory analyzes.

**(E) Depth-dependent robustness (Reviewer Jyme).**
We clarified that the trend—attenuation of sensitivity with depth—follows directly from the recurrence relation in our theory, which predicts increased cancellation across layers. The magnitude of this attenuation is inherently input-dependent and is therefore quantified empirically through per-layer local sensitivity and gradient alignment (Figures 4 and 5).

---

### 3. Summary of Reviewer Positions

- **Reviewer NCE2 (score 4)**: Several concerns stemmed from scope misunderstanding (e.g., expecting comparison to robustness-oriented attention variants or heatmap analysis). These were clarified in the rebuttal.

- **Reviewer da2g (score 6)**: Strongly positive. Requested clarification on local linearity and stronger attacks. Addressed through explanations and new experiments (large $\epsilon$, restarts).

- **Reviewer Jyme (score 6)**: Positive overall. Questions on depth-dependent robustness and the attenuation factor were resolved with expanded explanations and quantitative results.

- **Reviewer kz14 (score 6)**: Positive but raised concerns about dataset scale and attack diversity. We clarified the role of controlled small-scale studies and justified the use of standard white-box attacks.

---

We appreciate your time and consideration in assessing our submission.

---

### Meta-Review · Area_Chair_GcAK · 2025-12-29

**Summary:**

The main concerns about this paper lie in the technical as well as the experimental perspectives.

1. Overreliance on local linear assumptions in theoretical analysis (gradient and Lipschitz constants) .
2. Vision-only experiments are conducted on small, low-resolution datasets (CIFAR, Tiny-ImageNet) instead of more realistic, high-resolution datasets like ImageNet-1k.
3. The evaluation omits transformer-aware and more complex attack methods, and uses limited attack budgets (e.g., PGD/CW) without testing stronger ones.
4. Weak VLM Attack Protocol: The protocol for attacking CLIP/DiffCLIP is too basic; stronger, field-standard multi-prompt and multimodal attack methods should be used.
5. Lack of comparative baselines and limited gradient and perturbation analyses.
6. Limited robustness evaluation and lack of mechanistic explanation.
7. Lack detailed visualizations or quantitative examples, like attention maps, gradient heatmaps, or perturbation trajectories.

**Reviewer Concerns:**

As the authors pointed out, they provided a full consideration of the scope of comparison (Reviewers NCE2, kz14),  local linear assumptions (Reviewer da2g), stronger adversarial settings, transformer-aware attacks (Reviewer kz14), and depth-dependent robustness (Reviewer Jyme). All the questions were clarified in the rebuttal materials.

**Reviewer Scores:**

The original scores of this paper are 6,6,6,4. After reading the rebuttal, the AC thinks the authors have almost addressed these concerns from the technical and experimental parts.

---

### Decision · Program_Chairs · 2026-01-26

Accept (Poster)